# On the Gradient Formula for learning Generative Models with Regularized Optimal Transport Costs

**Antoine Houdard**      *Antoine.Houdard@math.u-bordeaux.fr*
**Arthur Leclaire**      *Arthur.Leclaire@math.u-bordeaux.fr*
**Nicolas Papadakis**      *Nicolas.Papadakis@math.u-bordeaux.fr*
*Univ. Bordeaux, Bordeaux INP, CNRS, IMB, UMR 5251,F-33400 Talence, France*

**Julien Rabin**      *Julien.Rabin@unicaen.fr*
*Normandie Univ., UNICAEN, ENSICAEN, CNRS, GREYC, 14000 Caen, France*

**Reviewed on OpenReview:** *https://openreview.net/forum?id=FbztvhdCX9*

## Abstract

Learning a Wasserstein Generative Adversarial Networks (WGAN) requires the differentiation of the optimal transport cost with respect to the parameters of the generative model. In this work, we provide sufficient conditions for the existence of a gradient formula in two different frameworks: the case of semi-discrete optimal transport (*i.e.* with a discrete target distribution) and the case of regularized optimal transport (*i.e.* with an entropic penalty). In both cases the gradient formula involves a solution of the semi-dual formulation of the optimal transport cost. Our study makes a connection between the gradient of the WGAN loss function and the Laguerre diagrams associated to semi-discrete transport maps. The learning problem is addressed with an alternating algorithm, which is in general not convergent. However, in most cases, it stabilizes close to a relevant solution for the generative learning problem. We also show that entropic regularization can improve the convergence speed but noticeably changes the shape of the learned generative model.

## 1 Introduction

In data science, generative models are ubiquitous to approximate a data distribution in order to draw new samples, to interpolate data points, to design a geometric structure revealing some interpretable dimensions of data (Radford et al., 2016; Shen et al., 2020) or to restore corrupted data by regularizing inverse problems (Bora et al., 2017; Hand & Joshi, 2019; Hyder & Asif, 2020; Heckel & Soltanolkotabi, 2020; Menon et al., 2020; Shamshad & Ahmed, 2020; Damara et al., 2021; Leong, 2021). Given the empirical distribution $\nu$ of the data supported on a compact set $\mathcal{Y} \subset \mathbb{R}^d$, estimating a generative network consists in solving

$$\arg\min_{\theta \in \Theta} \mathcal{L}(\mu_\theta, \nu) \tag{1}$$

where $\mu_\theta$ is the distribution of generated samples (parameterized by a $\theta$ in an open subset $\Theta \subset \mathbb{R}^q$) with support included in a compact set $\mathcal{X} \subset \mathbb{R}^d$, and where $\mathcal{L}$ is a loss function between probability distributions. The distribution $\mu_\theta$ is often considered to be the law of a random variable $g_\theta(Z)$ where $g_\theta$ is a neural network and $Z$ a random variable, and samples of the model can be obtained by passing new realizations of $Z$ through the network $g_\theta$. These distributions are generally built upon features computed from samples and data points, such as the latent space of a variational auto-encoder (Kingma & Welling, 2014).

The choice of the loss function is crucial to quantify the discrepancy between probability distributions. In the seminal work of Goodfellow et al. (2014) on generative adversarial networks (GAN), the loss function (1) is related, in a dual sense, to the Jensen-Shannon divergence between feature distributions. The Jensen-Shannon divergence is nevertheless not robust enough to data outliers and leads to convergence issues when

learning the GAN model. For this reason, Arjovsky et al. (2017) later proposed to use optimal transport (OT) costs in the loss function, leading to new generative models called Wasserstein GANs (WGANs).

**Wasserstein distance**  The optimal transport cost between probability distributions $\mu_\theta$ and $\nu$ is

$$W(\mu_\theta, \nu) = \inf_{\pi \in \Pi(\mu_\theta, \nu)} \int_{\mathcal{X} \times \mathcal{Y}} c(x, y) d\pi(x, y), \tag{2}$$

where $\Pi(\mu_\theta, \nu)$ is the set of probability distributions on $\mathcal{X} \times \mathcal{Y}$ having marginals $\mu_\theta$ and $\nu$, while the ground cost $c : \mathcal{X} \times \mathcal{Y} \to \mathbb{R}$ is a continuous function in which $c(x, y)$ represents the elementary cost between locations $x \in \mathcal{X}$ and $y \in \mathcal{Y}$. A popular choice for the cost is the Euclidean distance to the power $p \geq 1$, *i.e.* $c(x, y) = \|x - y\|^p = \left( \sum_{i=1}^d (x_i - y_i)^2 \right)^{p/2}$, for which $W(., .)^{\frac{1}{p}}$ defines the well-known $p$-Wasserstein distance. The cost function may also be defined as a metric in feature space. As we will recall later, the OT cost (2) admits a dual formulation (Santambrogio, 2015)

$$W(\mu_\theta, \nu) = \sup_{(\varphi, \psi) \in \mathcal{K}_c} \int \varphi d\mu_\theta + \int \psi d\nu, \tag{3}$$

where $(\varphi, \psi)$ is a couple of dual variables that belongs to the set

$$\mathcal{K}_c = \{ (\varphi, \psi) \in \mathscr{C}(\mathcal{X}) \times \mathscr{C}(\mathcal{Y}), \text{ subject to } \varphi(x) + \psi(y) \leq c(x, y) \quad \mu \otimes \nu \text{ a.e. } \} \tag{4}$$

where $\mathscr{C}(\mathcal{X})$ indicates the set of real continuous functions on $\mathcal{X}$ and $\otimes$ the product of two measures. Optimizing one of the dual variables in (3) amounts to taking the $c$-transform

$$\psi^c(x) = \min_{y \in \mathcal{Y}} c(x, y) - \psi(y), \tag{5}$$

thus leading to another expression of the OT cost, called "semi-dual formulation":

$$W(\mu_\theta, \nu) = \sup_{\psi \in \mathscr{C}(\mathcal{Y})} \int \psi^c d\mu_\theta + \int \psi d\nu. \tag{6}$$

Solving the constrained dual problem (3) is a difficult and possibly infinite-dimensional optimization problem. To circumvent this issue, one can turn to the entropic formulation of OT (Chizat, 2017; Peyré & Cuturi, 2019) which is an unconstrained optimization problem on $(\varphi, \psi)$, where the regularization of the OT problem is controlled by a parameter $\lambda > 0$. The entropy-regularized OT admits a similar semi-dual formulation (6) except that a softmin (a smoothed version of the minimum) is involved in the computation of the $c$-transform (5). In the discrete setting, the regularized OT cost can be computed efficiently with the Sinkhorn algorithm (Peyré & Cuturi, 2019), which exhibits geometric convergence. When one of the distribution is continuous (as in (1)), one has to rely on stochastic algorithms (Genevay et al., 2016), which are provably convergent, but considerably slower.

**Learning Wasserstein Generative models**  In this paper we study the case where the loss function $\mathcal{L}$ for learning the generative model is the OT cost with entropic regularization $\lambda$ (including the unregularized case $\lambda = 0$, see Definition 2 below):

$$\arg \min_{\theta \in \Theta} W_\lambda(\mu_\theta, \nu). \tag{7}$$

In practice, the optimization problem (7) can be solved with a gradient-based algorithm, such as the ADAM algorithm (Kingma & Ba, 2015), that requires the computation of the gradients of the Wasserstein cost with respect to $\theta$. To compute this gradient, we rely on the semi-dual formulation of regularized OT. In particular, we are interested in the conditions ensuring that the gradient at a point $\theta_0$ satisfies

$$\nabla_\theta \big( W_\lambda(\mu_\theta, \nu) \big)_{|\theta = \theta_0} = \nabla_\theta \left( \int \varphi_* d\mu_\theta \right)_{|\theta = \theta_0} \tag{8}$$

where $(\varphi_*, \psi_*)$ is an optimal dual variable for $W_\lambda(\mu_{\theta_0}, \nu)$. The objective of this paper is to provide a set of hypotheses that validate (8) and to show how it applies to generative models $g_\theta$ parameterized by neural networks.

## 1.1 Related works

Learning a generative model with a Wasserstein cost requires the computation of the gradient (8), which involves the optimal dual variable $\varphi_*$. We now detail related works on differentiation of OT costs, learning of generative models with OT costs, dual OT solver and semi-discrete OT.

**Differentiability of OT costs**  The gradient formula (8) is given in the seminal paper on WGAN (Arjovsky et al., 2017) with the assumption that both sides of the equality exist. This formula has been extended to more general costs and exploited in several papers related to WGAN learning, for example (Liu et al., 2019). In the context of discrete regularized OT, one can find in (Sanjabi et al., 2018, Appendix C) a gradient formula expressed through the primal formulation, with a short proof limited to discrete regularized OT. All these results are based on some version of the so-called "envelope theorem" (also called Danskin's theorem in the context of convex optimization), which allows to differentiate under the maximum. This theorem also requires regularity assumptions on the generator $g_\theta$. As we will see in Section 2.3, in the discrete setting, there exist some irregular cases where these assumptions are not sufficient to make formula (8) licit.

In parallel to the literature on WGAN, the regularity and gradient of regularized OT costs $p \mapsto W_\lambda(p, q)$ (with $\lambda > 0$) are studied in (Cuturi & Peyré, 2016, Prop. 2.3) for discrete distributions $(p, q)$. The Gâteaux-differentiability of $(p, q) \mapsto W_\lambda(p, q)$ is also proved in (Feydy et al., 2019), and extended to continuous differentiability in (Bigot et al., 2019, Prop. 2.3). From these results, if $(p_\theta)$ is a parametric family of distribution supported on the *same* finite set, the chain rule gives the differentiability of $\theta \mapsto W_\lambda(p_\theta, q)$.

**Wasserstein Generative Models**  The method of Arjovsky et al. (2017) is based on the 1-Wasserstein distance and it only requires one dual variable that is constrained to be 1-Lipschitz. In practice, this dual variable is parameterized by a neural network, and the Lipschitz constraint is enforced by weight clipping (WGAN-WC). On a similar formulation, Gulrajani et al. (2017) encourage Lipschitzness by including a gradient penalty in the dual loss (WGAN-GP). Genevay et al. (2018) introduce the so-called Sinkhorn divergences (based on regularized OT) which reduce the problem of sample complexity when learning generative models. Seguy et al. (2018) also consider regularized OT costs with a generic cost function. This leads to an unconstrained dual problem, but with two dual variables. In practice, the authors choose to parameterize both dual variables with neural networks. For the same problem, the authors of Sanjabi et al. (2018) study the convergence and stability of the training procedure, the convergence being proved for the case of discrete distributions. In particular, they give the expression of the gradient of (8), under a primal form, in the case of discrete regularized OT. This gradient expression is exploited to perform WGAN training by stochastic gradient descent. In Liu et al. (2019), a similar regularized OT framework is applied to empirical distributions (*i.e.* discrete distributions obtained from samples) to learn a generator, with the particularity that the cost function depends on a set of simultaneously-learned features.

Closer to our framework, Chen et al. (2019) consider the semi-dual formulation of OT (6). In this work, the dual variable $\psi$ is optimized with the the Average Stochastic Gradient Descent (ASGD) algorithm proposed in (Genevay et al., 2016). Contrary to (Seguy et al., 2018), the dual variable $\psi$ is not parameterized with a neural network, which hinders the applicability of the method to very large datasets. Indeed, as shown in (Leclaire & Rabin, 2021), the convergence speed of ASGD for optimizing $\psi$ decreases when either the dimension $d$ of samples or the dimension of vector $\psi$ (equal to the number of training points) increases. In (Chen et al., 2019), the primal solution $\pi$ of (2) is finally used to perform a gradient step on the generative model. As a result, their whole algorithm is not expressed as a direct minimization of (1). This is a main difference with our work, in which the proposed gradient calculation for (8) makes it possible to train the generator $g_\theta$ and the dual variable $\psi$ with an alternating min-max procedure on the dual cost (3).

In a similar work, Mallasto et al. (2019a) parameterize the dual variable $\psi$ by a neural network and obtain the second dual variable $\varphi$ with an approximated $c$-transform computed on mini-batches. With this approximation, the pair of dual variables may not satisfy the constraint (4). A penalty on $c(x, y) - \varphi(x) - \psi(y)$ is thus integrated in the loss function. This approach scales up to large databases, with a relatively accurate estimation of the Wasserstein cost. On batch strategy, Fatras et al. (2020) consider an alternative Wasserstein cost defined as an expectation over mini-batches. This stochastic approximation introduces an estimation bias which is shown in practice to regularize the transportation problem.

**OT dual solvers** Several authors (Mallasto et al., 2019b; Stanczuk et al., 2021; Korotin et al., 2021; 2022) question the performance of OT dual solvers and its impact on WGAN learning. Mallasto et al. (2019b) compare several algorithms to estimate the dual variables, namely WGAN-WC, WGAN-GP and two methods denoted as "$c$-transform" and "$(c, \varepsilon)$-transform". These two last methods, while better estimating the OT cost, do not improve the visual quality of the generative model when used in WGAN learning, producing very blurry images for the CelebA database. These findings are confirmed by Stanczuk et al. (2021) who provide explanations related to biased estimation of Wasserstein gradients, false Wasserstein minima supported on barycenters, and limitations of the use of the Euclidean distance between natural images.

To pursue these investigations, benchmarks for OT solvers are proposed in (Korotin et al., 2021; 2022). In these works, absolutely continuous measures $\mu, \nu$ together with their corresponding groundtruth OT map are provided for the 2-Wasserstein and 1-Wasserstein costs. The compared OT solvers are all based on parameterizations of the dual problem with neural networks, whose optimization requires to draw some batches of $\mu$ and $\nu$, which regularizes the OT cost and induces an estimation bias.

In this paper, in order to stay as close as possible to the original OT cost, we do not apply a batch procedure on $\nu$, thus restricting our experiments to smaller datasets. Our numerical study shows that, with the simpler MNIST database, a WGAN learning based on $c$-transforms computed on all datapoints (*i.e.* not batchwise) can produce sharp images while relying on a provably-convergent estimation algorithm for the Wasserstein cost. Also, our theoretical analysis of gradients enables to better understand the impact of the errors made by the OT dual solver: the performance of the dual solver does not guarantee the sharpness of generated images, but it plays a role in making the generative distribution cover the whole database.

**Semi-discrete OT** The theoretical results of the present paper rely heavily on concepts related to semi-discrete OT, that is, OT between an absolutely continuous distribution $\mu$ and a discrete distribution $\nu$ supported on a finite set. The connection between semi-discrete OT and a concave optimization problem is originally made in Aurenhammer et al. (1998). Many numerical solutions have been proposed to solve semi-discrete OT, see Kitagawa et al. (2017) and references therein. From these works, we will use the concepts of Laguerre diagram (illustrated in Fig. 2) and $c$-transforms of functions defined on a finite set. The concept of $c$-transform is also central in the construction of the benchmark of (Korotin et al., 2022), where the so-called "MinFunnel" functions can be seen, for $c(x, y) = \|x - y\|$, as $c$-transforms attached to a set of discrete points. Finally, let us mention that Lei et al. (2019) studied some connections between WGAN learning and several formulations of the semi-discrete OT problem, recalling its equivalence with other well-known problems from convex geometry. In this perspective, the present paper provides new insights on the connection between the WGAN and OT problems, which pursue the observations formulated in (Genevay et al., 2017).

## 1.2 Contributions and outline

The main purpose of this paper is to propose a complete set of hypotheses that ensure the validity of the gradient formula referred to as equation (8) above and (Grad-OT) in a more general form below. Our approach is not restricted to the cost $c(x, y) = \|x - y\|$ related to the 1-Wasserstein distance. It involves weak regularity hypotheses on the cost (called $x$-regularity as introduced in Definition 1 below) and the generator (Definition 3). Based on this gradient formula, we consider a stochastic algorithm for learning a generative model that can be understood as an alternating optimization algorithm on the semi-dual cost (6).

The main contributions of this paper are the following ones.

- In Section 2 we recall the WGAN learning framework, and in particular necessary existing results on the dual formulations of OT costs. Then we provide a counterexample (Proposition 2) that illustrates how (Grad-OT) can fail for unregularized OT when a technical hypothesis is not met.

- Section 3 focuses on the gradient formula (Grad-OT) in the case of unregularized ($\lambda = 0$) semi-discrete OT (*i.e.* when $\nu$ is discrete). We first prove the formula for $x$-regular costs (Theorem 3) and extend it to less regular costs (Theorem 4). Both these results are based on a technical assumption that for any $\theta$, $\mu_\theta$ does not charge a particular subset called the Laguerre interface.

- In Section 4, we extend the gradient formula to the case of entropy-regularized OT, proving (Grad-OT) for $x$-regular costs (Theorem 5) with no assumption on $\nu$. We also adapt this result for the differentiation of the Sinkhorn divergence (Theorem 6).

- In Section 5, we illustrate the behavior of the alternating optimization algorithm for generative model learning. First, with the toy example of Proposition 2, we examine the trajectory of the optimization algorithm, and explain why it does not get trapped in a singular point with no gradient. Second, for various synthetic cases in dimension 2, we exhibit that the alternating algorithm generally stabilizes, but may also oscillate or converge to sub-optimal configurations, depending on the choice of gradient step size strategies. Third, we consider the learning of a WGAN for MNIST digits generation. In this high-dimensional case, we examine the stability of the loss function, and underline the impact of the entropic regularization both on numerical and visual results.

We emphasize that the theorems ensuring (Grad-OT) are based on weak regularity assumptions for the cost and the generator that can be met in practice. Our regularity hypotheses are for instance true with the quadratic cost $c(x, y) = \|x - y\|^2$ and with generators given as neural networks with $\mathscr{C}^1$ activation functions. Also, in the case of unregularized OT, the technical hypothesis is true except for degenerate positioning of the support of $\mu_\theta$. Therefore, our results apply to most practical cases of WGAN training, and explain why the algorithms for WGAN learning generally do not get caught in singular points with inexisting gradients.

## 2 The Wasserstein GAN Problem

In this section, we introduce in Section 2.1 the primal, dual and semi-dual formulations of the OT cost. The OT problem is presented in the general case including entropic regularization with parameter $\lambda \geq 0$, thus encompassing the unregularized case $\lambda = 0$. Next, we introduce in Section 2.2 the generative learning problem as well as the regularity hypothesis on the generator, which will be used to show the existence of gradients of the OT cost with respect to $\theta$. In Section 2.3, we finally present a counter-example of measures $\mu_\theta$ and $\nu$ for which the gradient formula (8) does not hold.

First, let us give some notations that are used in the whole paper. Let $\mathcal{X}, \mathcal{Y}$ be compact subsets of $\mathbb{R}^d$, and let $c : \mathcal{X} \times \mathcal{Y} \to \mathbb{R}$ be a continuous function. Let $\Theta$ be an open subset of $\mathbb{R}^q$ used to parameterize the generator $g_\theta$. We say that $\varphi : \mathcal{X} \to \mathbb{R}$ is $\mathscr{C}^1$ on $\mathcal{X}$ if it is $\mathscr{C}^1$ on an open neighborhood of $\mathcal{X}$.

**Definition 1** ($x$-regularity). *We say that $c$ is $x$-**regular** if there is $L > 0$ and an open set $U$ with $\mathcal{X} \subset U \subset \mathbb{R}^d$ such that for any $y \in \mathcal{Y}$, $c(\cdot, y)$ can be extended to a $L$-Lipschitz $\mathscr{C}^1$ function on $U$.*

We illustrate this definition with the cost $c(x, y) = \|x - y\|^p$ on $\mathbb{R}^d$, with $p \geq 1$. For $p > 1$, $c$ is a smooth function on $\mathbb{R}^d \times \mathbb{R}^d$ and thus $x$-regular, as $\nabla_x c$ is a continuous function on $\mathbb{R}^d \times \mathbb{R}^d$ that is bounded on $U \times \mathcal{Y}$ for any bounded open set $U$ containing $\mathcal{X}$. For $p = 1$, the cost $c(x, y) = \|x - y\|$ is not $x$-regular as soon as $\mathcal{X}$ and $\mathcal{Y}$ are not disjoint. Indeed, for a fixed $y \in \mathcal{Y}$, $x \mapsto \|x - y\|$ is not differentiable at $y$.

### 2.1 Optimal Transport, Primal and Dual Problems

**Definition 2** (Primal formulation). *Let $\mu, \nu$ be two probability measures on $\mathcal{X}, \mathcal{Y}$ respectively. For $\lambda \geqslant 0$, the regularized optimal transport cost is defined by*

$$W_\lambda(\mu, \nu) = \inf_{\pi \in \Pi(\mu, \nu)} \int_{\mathcal{X} \times \mathcal{Y}} c \, d\pi + \lambda \, \mathrm{KL}(\pi | \mu \otimes \nu) \tag{9}$$

*where $\Pi(\mu, \nu)$ is the set of probability distributions on $\mathcal{X} \times \mathcal{Y}$ with marginals $\mu$ and $\nu$, and where $\mathrm{KL}(\pi | \mu \otimes \nu) = \int \log(\frac{d\pi}{d\mu \otimes \nu}) d\pi$ if $\pi$ admits a density $\frac{d\pi}{d\mu \otimes \nu}$ w.r.t. $\mu \otimes \nu$ and $+\infty$ otherwise.*

**Theorem 1** (Dual formulation (Santambrogio, 2015; Genevay, 2019; Feydy et al., 2019)). *Strong duality holds in the sense that*

$$W_\lambda(\mu, \nu) = \max_{\varphi \in \mathscr{C}(\mathcal{X}), \psi \in \mathscr{C}(\mathcal{Y})} \int \varphi(x) d\mu(x) + \int \psi(y) d\nu(y) - \int m_\lambda\big(\varphi(x) + \psi(y) - c(x, y)\big) d\mu(x) d\nu(y) \tag{10}$$

*where, for $\lambda = 0$, $m_0(t) = 0$ if $t \geq 0$, and $+\infty$ otherwise, and for $\lambda > 0$, $m_\lambda(t) = \lambda(e^{\frac{t}{\lambda}} - 1)$ A solution $(\varphi, \psi)$ of this dual problem is called a pair of Kantorovich potentials. When $\lambda > 0$, the solutions are uniquely defined almost everywhere up to an additive constant (i.e. if $(\varphi, \psi)$ is a solution, then any solution can be written $(\varphi - k, \psi + k)$ with $k \in \mathbb{R}$). Also, when $\lambda > 0$, the primal problem admits a unique solution*

$$d\pi(x, y) = \exp\left(\frac{\varphi(x) + \psi(y) - c(x, y)}{\lambda}\right) d\mu(x) d\nu(y). \tag{11}$$

If $\lambda = 0$ the primal solution $\pi$ of (9) may not be absolutely continuous w.r.t. $\mu \otimes \nu$ anymore. Nevertheless, under weak assumptions (e.g. $\mu, \nu$ admit second-order moments and $\mu$ is absolutely continuous), one can show (Santambrogio, 2015) that the support of $\pi$ is actually supported on the graph of an optimal transport map $T^*$, i.e. the optimal $\pi$ is the probability distribution of $(X, T^*(X))$ where $X$ has distribution $\mu$ and $T^*(X)$ has distribution $\nu$.

For $\psi \in \mathscr{C}(\mathcal{Y})$, let us define the regularized $c$-transform as in (Feydy et al., 2019)

$$\forall x \in \mathcal{X}, \quad \psi^{c,\lambda}(x) = \operatorname*{softmin}_{y \in \mathcal{Y}} c(x, y) - \psi(y) \tag{12}$$

where the softmin operation is defined as

$$\operatorname*{softmin}_{y \in \mathcal{Y}} u(y) = \begin{cases} \min_{y \in \mathcal{Y}} u(y) & \text{if } \lambda = 0, \\ -\lambda \log \int e^{-\frac{u(y)}{\lambda}} d\nu(y) & \text{if } \lambda > 0. \end{cases} \tag{13}$$

We also define the analoguous operators for the $x$-variable. It must be noted that the regularized $c$-transform $\psi^{c,\lambda}$ also depends on $\nu$ even if $\nu$ is omitted in the notation.

Given a pair of dual variables $(\varphi, \psi) \in \mathscr{C}(\mathcal{X}) \times \mathscr{C}(\mathcal{Y})$, one can see that taking $c$-transforms

$$\begin{cases} \tilde{\psi} = \varphi^{c,\lambda} \\ \tilde{\varphi} = \psi^{c,\lambda} \end{cases}, \tag{14}$$

leads to a new pair $(\tilde{\varphi}, \tilde{\psi})$ of dual variables that have a better dual cost (10) than $(\varphi, \psi)$. Thus the dual problem can be restricted to $c$-concave functions, i.e. functions that write as $c$-transforms. Such $c$-concave functions inherits regularity from the cost function (see Appendix C) and can be extrapolated to any $x \in \mathcal{X}$.

**Theorem 2** (Semi-dual formulation (Genevay, 2019)). *The dual problem (10) is equivalent to the semi-dual problem*

$$W_\lambda(\mu, \nu) = \sup_{\psi \in \mathscr{C}(\mathcal{Y})} \int \psi^{c,\lambda}(x) d\mu(x) + \int \psi(y) d\nu(y). \tag{15}$$

*A solution $\psi$ of the semi-dual problem is called a Kantorovich potential. In other words, $\psi$ is a Kantorovich potential if and only if $(\psi^{c,\lambda}, \psi)$ is a pair of Kantorovich potentials. By symmetry, we can also formulate a semi-dual problem on the dual variable $\varphi$.*

## 2.2 Learning a Generative Network

Estimating a Wasserstein generative adversarial network (WGAN) from an empirical data distribution $\nu$ consists in minimizing

$$h_\lambda(\theta) = W_\lambda(\mu_\theta, \nu), \tag{16}$$

where the generated distribution $\mu_\theta$ is assumed to be the distribution of $g_\theta(Z)$, with $Z$ a random variable.

Denoting by $\zeta$ the probability distribution of $Z$ on the measurable space $\mathcal{Z}$, $\mu_\theta$ is the image measure of $\zeta$ by the generator $g_\theta$, also known as the pushforward $\mu_\theta = g_\theta \sharp \zeta$. More precisely, the notation $g_\theta$ refers to $g(\theta, \cdot)$ where $g : \Theta \times \mathcal{Z} \to \mathbb{R}^d$ is a function defined on the product of the open set $\Theta \subset \mathbb{R}^q$ with $\mathcal{Z}$. In the following, we give different sets of conditions on $g$ that allow to compute the derivatives of $h_\lambda$.

All the results of this paper are related to the behavior of the function

$$I(\varphi, \theta) = \int_{\mathcal{X}} \varphi d\mu_\theta, \quad (\varphi \in \mathscr{C}(\mathcal{X}), \ \theta \in \Theta), \tag{17}$$

which allows to reformulate the semi-dual expression of optimal transport (15) as

$$h_\lambda(\theta) = \max_{\psi \in \mathscr{C}(\mathcal{Y})} I(\psi^{c,\lambda}, \theta) + \int_{\mathcal{Y}} \psi d\nu. \tag{18}$$

In order to study the gradient of $h_\lambda$, we define $F_\lambda : \mathscr{C}(\mathcal{Y}) \times \Theta \to \mathbb{R}$ with

$$\forall \psi \in \mathscr{C}(\mathcal{Y}), \forall \theta \in \Theta, \quad F_\lambda(\psi, \theta) = I(\psi^{c,\lambda}, \theta) = \int_{\mathcal{X}} \psi^{c,\lambda} d\mu_\theta = \mathbb{E}[\psi^{c,\lambda}(g_\theta(Z))], \tag{19}$$

where the expectation is taken with respect to the probability distribution $\zeta$ of $Z$.

Combining all previous definitions, the problem we tackle writes

$$W_\lambda(\mu_\theta, \nu) = h_\lambda(\theta) = \max_{\psi \in \mathscr{C}(\mathcal{Y})} H_\lambda(\psi, \theta) := F_\lambda(\psi, \theta) + \int_{\mathcal{Y}} \psi d\nu, \tag{20}$$

and our objective is to study the relationship between $\nabla h_\lambda(\theta)$ and $\nabla_\theta F_\lambda(\psi, \theta)$. In the unregularized case, we use simpler notations, $h$, $W$, $H$, and $F$ for $h_0$, $W_0$, $H_0$ and $F_0$ respectively, dropping the index $\lambda$.

As we will see later, computing the derivatives of $h_\lambda$ boils down to differentiating under the max, which is allowed by the so-called envelope theorem. This result appears under different forms in the literature (Oyama & Takenawa, 2018). In Appendix D, we recall the version of the envelope theorem that is used in our proofs.

If we admit the differentiability of all terms, the computation of the gradient is straightforward:

**Proposition 1.** *Let $\theta_0$ and $\psi_0$ satisfying $h_\lambda(\theta_0) = H_\lambda(\psi_0, \theta_0)$. If $h_\lambda$ and $\theta \mapsto F_\lambda(\psi_0, \theta)$ are both differentiable at $\theta_0$, then*

$$\nabla h_\lambda(\theta_0) = \nabla_\theta F_\lambda(\psi_0, \theta_0). \tag{Grad-OT}$$

*Proof.* First, notice that $H_\lambda(\psi_0, \cdot)$ and $F_\lambda(\psi_0, \cdot)$ differ by a constant, and thus have same gradients. Using (18), $h_\lambda(\theta) \geqslant H_\lambda(\psi_0, \theta)$ for any $\theta$, with equality if $\theta = \theta_0$. Therefore, the function $\theta \mapsto h_\lambda(\theta) - H_\lambda(\psi_0, \theta)$ has a minimum at $\theta = \theta_0$, and its gradient at $\theta_0$ vanishes, so that $\nabla h_\lambda(\theta_0) = \nabla_\theta H_\lambda(\psi_0, \theta_0)$. $\square$

However, showing the existence of $\nabla_\theta F_\lambda(\psi_0, \theta_0)$ is not straightforward, as one has to differentiate under the expectation in (19). To this end, we introduce the following technical hypothesis.

**Definition 3** (Hypothesis $(\mathsf{G}_\Theta)$). *For $\theta_0 \in \Theta$, we say that $g : \Theta \times \mathcal{Z} \to \mathcal{X}$ satisfies Hypothesis $(\mathsf{G}_{\theta_0})$ if there exists a neighborhood $V$ of $\theta_0$ and $K \in L^1(\zeta)$ such that almost surely $\theta \mapsto g(\theta, Z)$ is $\mathscr{C}^1$ on $V$ with differential $\theta \mapsto D_\theta g(\theta, Z)$ and*

$$\forall \theta \in V, \quad \zeta\text{-a.s.} \quad \|g(\theta, Z) - g(\theta_0, Z)\| \leqslant K(Z)\|\theta - \theta_0\|. \tag{21}$$

*We say that $g$ satisfies Hypothesis $(\mathsf{G}_\Theta)$ if $g$ satisfies Hypothesis $(\mathsf{G}_{\theta_0})$ for any $\theta_0 \in \Theta$.*

A sufficient condition for Hypothesis $(\mathsf{G}_\Theta)$ is that almost surely, $\theta \mapsto g(\theta, Z)$ is $\mathscr{C}^1$ on $\Theta$ and that there exists $K \in L^1(\zeta)$ such that

$$\forall \theta, \theta' \in \Theta, \quad \zeta\text{-a.s.} \quad \|g(\theta', Z) - g(\theta, Z)\| \leqslant K(Z)\|\theta' - \theta\|. \tag{22}$$

Hypothesis $(\mathsf{G}_\Theta)$ is also satisfied as soon as $g$ is $\mathscr{C}^1$ on $\Theta \times \mathcal{Z}$ with $\mathcal{Z} \subset \mathbb{R}^s$ compact. In this case, for any $\theta_0 \in \Theta$, there is $r > 0$ such that $V = \overline{B}(\theta_0, r)$ is included in $\Theta$ and then (21) holds with the constant bound $K = \sup_{\Theta \times \mathcal{Z}} \|D_\theta g\|$. This is the case when $g$ is parameterized by a neural network with $\mathscr{C}^1$ activation functions, and with an input noise $Z$ supported on a bounded subset of $\mathbb{R}^s$ (*e.g.* $Z$ uniform on $[-1, 1]^s$). In Appendix A, we show that Hypothesis $(\mathsf{G}_\theta)$ is satisfied in a more general setting with integrable noise.

### 2.3 A telling counter-example

We now show that the differentiation with respect to $\theta$ under the max in (18) can fail, even for regular generators $g$, thus discarding the formula (Grad-OT). To illustrate this point, let us consider a simple unregularized OT problem (*i.e.* $\lambda = 0$) between a Dirac $\delta_\theta$ located at $\theta \in \mathbb{R}^d$ and a sum of two Diracs at positions $y_1 \neq y_2 \in \mathbb{R}^d$. This setting can be obtained by setting $g_\theta(z) = \theta$ (for any distribution of $Z$).

**Proposition 2.** *Let $\mu_\theta = \delta_\theta$ and $\nu = \frac{1}{2}\delta_{y_1} + \frac{1}{2}\delta_{y_2}$.*

*For $p \geq 1$, consider the cost $c(x,y) = \|x - y\|^p$ where $\|\cdot\|$ is the Euclidean norm on $\mathbb{R}^d$. Then*

- *$h(\theta) = W(\mu_\theta, \nu)$ is differentiable at any $\theta \notin \{y_1, y_2\}$ for $p = 1$, and at any $\theta$ for $p > 1$,*

- *for any $\psi_{*0} \in \arg\max_\psi H(\psi, \theta_0)$, the function $\theta \mapsto F(\psi_{*0}, \theta)$ is **not** differentiable at $\theta_0$.*

*Hence relation (Grad-OT) never stands.*

*Proof.* The only distribution on $\mathcal{X} \times \mathcal{Y}$ having marginals $\mu_\theta, \nu$ is $\pi = \frac{1}{2}\delta_{(\theta, y_1)} + \frac{1}{2}\delta_{(\theta, y_2)}$. Therefore, recalling that $h(\theta) = W(\mu_\theta, \nu)$ and from the definition of the primal problem (9), we have

$$h(\theta) = \frac{1}{2}c(\theta, y_1) + \frac{1}{2}c(\theta, y_2) \tag{23}$$

which gives the first point.

Next, one can explicitly solve the dual problem, which, from the equivalent semi-dual formulation (15) and the definition (19) of $F$, reduces to the following optimization problem with respect to $(\psi(y_1), \psi(y_2)) \in \mathbb{R}^2$

$$\max_{\psi \in \mathbb{R}^2} H(\psi, \theta) \quad \text{with} \quad H(\psi, \theta) = \psi^c(\theta) + \frac{\psi(y_1) + \psi(y_2)}{2}, \tag{24}$$

where, for any $\psi$, the $c$-transform (13) writes

$$\psi^c(\theta) = \begin{cases} c(\theta, y_1) - \psi(y_1) & \text{if } c(\theta, y_1) - \psi(y_1) \leqslant c(\theta, y_2) - \psi(y_2), \\ c(\theta, y_2) - \psi(y_2) & \text{otherwise.} \end{cases} \tag{25}$$

Therefore,

$$H(\psi, \theta) = \begin{cases} c(\theta, y_1) + \frac{1}{2}(\psi(y_2) - \psi(y_1)) & \text{if } \psi(y_2) - \psi(y_1) \leqslant c(\theta, y_2) - c(\theta, y_1), \\ c(\theta, y_2) - \frac{1}{2}(\psi(y_2) - \psi(y_1)) & \text{otherwise.} \end{cases} \tag{26}$$

For a fixed $\theta_0$, $H(\cdot, \theta_0)$ is maximal at any $\psi_{*0}$ such that

$$\psi_{*0}(y_2) - \psi_{*0}(y_1) = c(\theta_0, y_2) - c(\theta_0, y_1). \tag{27}$$

Besides, from (26), one sees that $H(\psi, \cdot)$ is made of two pieces whose gradients can be computed explicitly for the cost $c(x, y) = \|x - y\|^p$:

$$\forall \theta \neq y_j, \quad \nabla_x c(\theta, y_j) = p\|\theta - y_j\|^{p-2}(\theta - y_j), \tag{28}$$

and, when $p > 1$ we also have $\nabla_x c(\theta, y_j) = 0$ for $\theta = y_j$. Therefore, as illustrated on Fig. 1, the gradients of the two pieces agree at no point $\theta$: for any $\theta$, $\nabla_x c(\theta, y_1) \neq \nabla_x c(\theta, y_2)$. In particular, the function $H(\psi, \cdot)$ is not differentiable at the interface $A_\psi = \{\, \theta \in \mathbb{R}^d \mid c(\theta, y_2) - c(\theta, y_1) = \psi(y_2) - \psi(y_1) \,\}$. As $F(\psi, \cdot)$ only differs from $H(\psi, \cdot)$ by the constant $\frac{\psi(y_1) + \psi(y_2)}{2}$, it is also not differentiable on $A_\psi$. But then, for $\psi_{*0}$ satisfying (27), $\theta_0$ lies on the interface $A_{\psi_{*0}}$, which gives the second point. Fig. 1 thus highlights the main pitfall in the proof: the measure $\delta_\theta$ puts some mass on a thin subset where $\theta \mapsto F(\psi_{*0}, \theta)$ is not smooth. $\qquad\square$

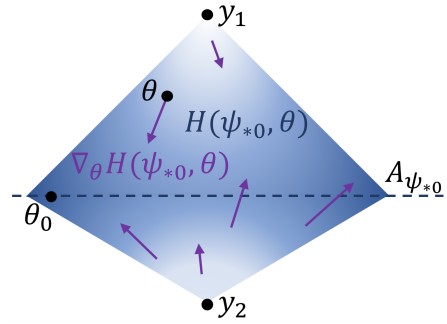

Figure 1: **Counter-example for the gradient formula.** This diagram illustrates the counter-example of Proposition 2 for $p = 2$. The notations used in this diagram are detailed in the proof. The function $\theta \mapsto H(\psi_{*0}, \theta)$ is drawn in blue background. For the optimal $\psi_{*0}$ corresponding to $\theta_0$, the function $\theta \mapsto H(\psi_{*0}, \theta)$ is not differentiable on the interface $A_{\psi_{*0}}$, where $\theta_0$ lies.

For the same example, we show in Appendix B that relation (Grad-OT) holds with regularized OT only in the case $p > 1$. This illustrates the need for a regularity hypothesis on the cost, even for regularized OT.

This counterexample can be extended to other situations where the measure $\mu_\theta$ is not supported on a single point. For example, for the same $\nu$, the gradient issue still happens if $\mu_\theta$ is the distribution of $\theta + Zu$ where $Z$ is a uniform random variable on $[-1, 1]$ and where $u \in \mathbb{R}^d$ is orthogonal to $y_2 - y_1$ ($\mu_\theta$ is the uniform distribution on a segment orthogonal to $[-1, 1]$ centred at $\theta$). In the following section, we adopt an hypothesis on the generator to avoid this objection.

## 3   Gradient formula in the unregularized semi-discrete setting

In this section, we prove the formula (Grad-OT) in the semi-discrete case, *i.e.* when the target measure $\nu$ is supported on a finite set of points. We only consider the unregularized case $\lambda = 0$ and recall that the index $\lambda$ is dropped in this setting, with the notations $W = W_0$, $h = h_0$, and $F = F_0$ respectively defined by (9), (16) and (19). More general results are given for regularized OT in the next section.

Our analysis builds on concepts related to semi-discrete OT (Aurenhammer et al., 1998; Kitagawa et al., 2017), and in particular to the notion of Laguerre diagram presented in Section 3.1. In Section 3.2, the Laguerre diagram is used to formulate a condition on the generator $g_\theta$ which allows the computation the gradient of $h$, by differentiating under the max in (20).

In the studied semi-discrete case, $\mathcal{X} \subset \mathbb{R}^d$ is compact and $\mathcal{Y}$ is finite with $J$ points, so that $\mathscr{C}(\mathcal{Y})$ identifies to $\mathbb{R}^J$. We recall that the objective is to show that $\nabla h(\cdot) = \nabla_\theta F(\psi, \cdot)$, so we study the function $F : \mathbb{R}^J \times \Theta \to \mathbb{R}$

$$F(\psi, \theta) = \int_{\mathcal{X}} \psi^c d\mu_\theta = \mathbb{E}[\psi^c(g_\theta(Z))]. \tag{29}$$

### 3.1   The Laguerre diagram and the semi-discrete transport maps

In this section, we recall the definitions of transport maps and Laguerre diagrams which are used in semi-discrete OT Aurenhammer et al. (1998); Kitagawa et al. (2017). We also give two lemmas in order to compute the gradient of $c$-transform functions. To any $\psi \in \mathbb{R}^J$, we associate a Laguerre diagram, which is a collection $(\mathrm{L}_\psi(y))_{y \in \mathcal{Y}}$ of Laguerre cells defined by

$$\mathrm{L}_\psi(y) = \{\, x \in \mathcal{X} \mid \forall y' \in \mathcal{Y} \setminus \{y\}, \; c(x, y) - \psi(y) < c(x, y') - \psi(y') \,\}. \tag{30}$$

We also define the Laguerre interface by

$$A_\psi = \mathcal{X} \setminus \bigcup_{y \in \mathcal{Y}} \mathrm{L}_\psi(y), \tag{31}$$

which corresponds to the set of points for which there is a "tie" in the minimum defining $\psi^c(x)$. For example, if $c(x, y) = \|x - y\|^2$ in $\mathbb{R}^d$, $A_\psi$ is contained in a union of hyperplanes whose directions are orthogonal to the segments $[y_1, y_2]$, $y_1, y_2 \in \mathcal{Y}$. In particular $A_\psi$ has zero Lebesgue measure, and thus the Laguerre diagram form a partition $\mathbb{R}^d$ up to a negligible set.

By construction, for $x \in \mathcal{X} \setminus A_\psi$, we can uniquely define the transport map

$$T_\psi(x) = \arg\min_{y \in \mathcal{Y}} c(x, y) - \psi(y). \tag{32}$$

If $\mu$ is a probability measure on $\mathcal{X}$, one can verify that for any $\psi \in \mathbb{R}^J$, $T_\psi$ is an optimal transport map between $\mu$ and $T_\psi \sharp \mu$. Conversely, if $\mu$ is absolutely continuous on $\mathcal{X}$, solving the OT problem from $\mu$ to $\nu$ is equivalent to find $\psi \in \mathbb{R}^J$ such that $T_\psi \sharp \mu = \nu$. We refer to Fig. 2 for an illustration of Laguerre diagram and semi-discrete transport maps; and to Kitagawa et al. (2017) for more details.

The two following lemmas will be useful to study the regularity of $F$ and $h$.

**Lemma 1.** *The map $(\psi, x) \mapsto T_\psi(x)$ is locally constant on $\mathbb{R}^J \times A_\psi$.*

*Proof.* Let $(\psi, x) \in \mathbb{R}^J \times A_\psi$ and let $y = T_\psi(x)$. By definition of Laguerre cells,

$$c(x, y) - \psi(y) - \min_{z \in \mathcal{Y} \setminus \{y\}} (c(x, z) - \psi(z)) < 0. \tag{33}$$

The left-hand side is a function that is jointly continuous in $(\psi, x)$ and is $< 0$ at $(\psi, x)$. Thus, there is a neighborhood $\mathsf{W}$ of $(\psi, x)$ where it stays negative, that is,

$$\forall (\psi', x') \in \mathsf{W}, \quad c(x', y) - \psi'(y) - \min_{z \in \mathcal{Y} \setminus \{y\}} (c(x', z) - \psi'(z)) < 0. \tag{34}$$

Therefore $T_{\psi'}(x') = y$ on $\mathsf{W}$. $\qquad \square$

**Lemma 2.** *Assume that $\mathcal{Y}$ is finite with $J$ points and that $c$ is $x$-regular (see Definition 1). Let $\psi \in \mathbb{R}^J$. Then $\psi^c$ is $\mathscr{C}^1$ on $\mathcal{X} \setminus A_\psi$ and*

$$\forall x \in \mathcal{X} \setminus A_\psi, \quad \nabla \psi^c(x) = \nabla_x c(x, T_\psi(x)). \tag{35}$$

*Proof.* First, one can notice that $\mathcal{X} \setminus A_\psi = \bigcup_{y \in \mathcal{Y}} \mathsf{L}_\psi(y)$ is an open subset of $\mathcal{X}$ because the Laguerre cells $\mathsf{L}_\psi(y)$ are open (thanks to the continuity of $c$). Besides, if $x \in \mathcal{X} \setminus A_\psi$, $y = T_\psi(x)$ is well-defined and $x \in \mathsf{L}_\psi(y)$. Thus $\mathsf{L}_\psi(y)$ is an open neighborhood of $x$ on which we have

$$\forall u \in \mathsf{L}_\psi(y), \quad \psi^c(u) = c(u, y) - \psi(y) \ .$$

Therefore, on $\mathsf{L}_\psi(y)$, $\psi^c$ is as regular as $c(\cdot, y) = c(\cdot, T_\psi(x))$ and has the same gradient. $\qquad \square$

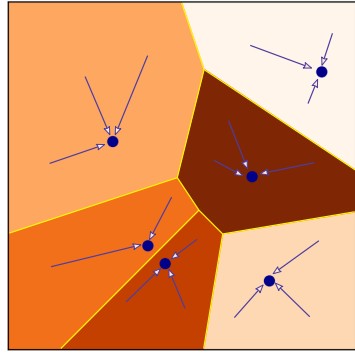 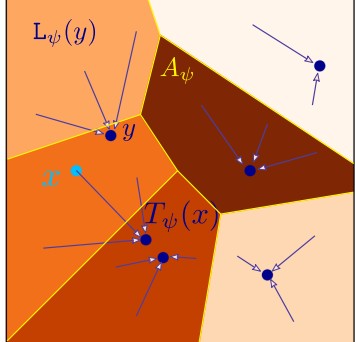

Voronoi diagram  Laguerre diagram

Figure 2: **Laguerre diagram and semi-discrete transport maps.** We display here two colored partitions corresponding to Laguerre diagrams associated to a set $\mathcal{Y}$ with 6 points (drawn in dark blue), and with the quadratic cost $c(x, y) = \|x - y\|^2$ on $[0, 1]^2$. On the left we display the Laguerre diagram for $\psi = 0$, which corresponds to the classical Voronoi diagram. On the right we display the Laguerre diagram for a specific value of $\psi$. The map $T_\psi$ is drawn as blue arrows. We also display in yellow the Laguerre interface $A_\psi$. Notice that, for a Laguerre diagram, a point $y$ does not necessarily belong to its Laguerre cell $\mathsf{L}_\psi(y)$.

### 3.2 Gradient of the unregularized Wasserstein loss function

We now state our main theorem for unregularized OT. To this end, we first analyse the regularity of $F$.

**Lemma 3** (Regularity of $F$). *Assume that $\mathcal{Y}$ is finite with $J$ points and that $c$ is $x$-regular. Assume that $g$ satisfies Hypothesis ($\mathsf{G}_{\theta_0}$) (Definition 3) at some $\theta_0 \in \Theta$. Assume that for any $\psi \in \mathbb{R}^J$, $\mu_{\theta_0}(A_\psi) = 0$, i.e. $g(\theta_0, Z) \in \mathcal{X} \setminus A_\psi$ almost surely. Then, for any $\psi \in \mathbb{R}^J$, $\theta \mapsto F(\psi, \theta)$ is differentiable at $\theta_0$ and, denoting as $D_\theta g$ the partial differential of $g$ with respect to $\theta$, we have*

$$\nabla_\theta F(\psi, \theta_0) = \mathbb{E}[D_\theta g(\theta_0, Z)^T \nabla \psi^c(g(\theta_0, Z))]. \tag{36}$$

*Proof.* Let $V$ be the neighborhood of $\theta_0$ given by Definition 3. For $\theta \in \Theta$, let us denote

$$f(\psi, \theta, Z) = \psi^c(g(\theta, Z)) \tag{37}$$

so that we get $F(\psi, \theta) = \mathbb{E}[f(\psi, \theta, Z)]$ from (29). The hypotheses on $g$ and Lemma 2 ensure that $f(\psi, \cdot, Z)$ is almost surely differentiable at $\theta_0$ and thanks to the chain-rule, we have

$$\nabla_\theta f(\psi, \theta_0, Z) := D_\theta g(\theta_0, Z)^T \nabla \psi^c(g(\theta_0, Z)). \tag{38}$$

Besides, since $c$ is $x$-regular, the $c$-transforms are $L$-Lipschitz thanks to Definition 1 and Lemma 6 given in Appendix C. Therefore, for any $\theta \in V$,

$$|f(\psi, \theta, Z) - f(\psi, \theta_0, Z)| \leqslant L\|g(\theta, Z) - g(\theta_0, Z)\| \leqslant LK(Z)\|\theta - \theta_0\| \quad \text{a.s.} \tag{39}$$

Next, replacing $\theta$ by $\theta_0 + t(\theta - \theta_0)$ for $t \in \mathbb{R}$ and letting $t \to 0$, we also get

$$\|\nabla_\theta f(\psi, \theta_0, Z).(\theta - \theta_0)\| \leqslant LK(Z)\|\theta - \theta_0\| \quad \text{a.s.} \tag{40}$$

In particular, $\mathbb{E}[\nabla_\theta f(\psi, \theta_0, Z)]$ exists. Therefore, we have for any $\theta \in V$,

$$\frac{|f(\psi, \theta, Z) - f(\psi, \theta_0, Z) - \nabla_\theta f(\psi, \theta_0, Z).(\theta - \theta_0)|}{\|\theta - \theta_0\|} \leqslant 2LK(Z) \quad \text{a.s.} \tag{41}$$

When $\theta \to \theta_0$, the left-hand-side tends almost surely to zero, and thus, the dominated convergence theorem ensures that

$$\mathbb{E}\left[\frac{|f(\psi, \theta, Z) - f(\psi, \theta_0, Z) - \nabla_\theta f(\psi, \theta_0, Z).(\theta - \theta_0)|}{\|\theta - \theta_0\|}\right] \longrightarrow 0 \tag{42}$$

and in particular,

$$F(\psi, \theta) - F(\psi, \theta_0) - \mathbb{E}[\nabla_\theta f(\psi, \theta_0, Z)].(\theta - \theta_0) = o(\|\theta - \theta_0\|). \tag{43}$$

This proves that $F(\psi, \cdot)$ is differentiable at $\theta_0$ with

$$\nabla_\theta F(\psi, \theta_0) = \mathbb{E}[\nabla_\theta f(\psi, \theta_0, Z)] = \mathbb{E}[D_\theta g(\theta_0, Z)^T \nabla \psi^c(g_{\theta_0}(Z))]. \tag{44}$$

$\square$

**Theorem 3.** *Assume that $\mathcal{Y}$ is finite with $J$ points and that $c$ is $x$-regular. Also assume that*

    *1. for any $\theta \in \Theta$, the Kantorovich potential $\psi_*$ for $W(\mu_\theta, \nu)$ (defined in Theorem 2) is unique up to additive constants.*

    *2. for any $\theta \in \Theta$ and any $\psi \in \mathbb{R}^J$, $\mu_\theta(A_\psi) = 0$, i.e. we have $g(\theta, Z) \in \mathcal{X} \setminus A_\psi$ a.s.*

    *3. $g$ satisfies Hypothesis ($\mathsf{G}_\Theta$) in Definition 3.*

*Then $h(\theta) = W(\mu_\theta, \nu)$ is differentiable at any $\theta \in \Theta$ and*

$$\nabla h(\theta) = \nabla_\theta F(\psi_*, \theta) = \mathbb{E}\left[D_\theta g(\theta, Z)^T \nabla \psi_*^c(g_\theta(Z))\right]. \tag{45}$$

*Proof.* The proof consists in applying the envelope Theorem 7 (recalled in Appendix D) to show that $h$ is differentiable at a fixed $\theta_0 \in \Theta$. Let us denote by $\psi_{*0}$ the corresponding Kantorovich potential for $W(\mu_{\theta_0}, \nu)$.

First, we need to build a selection of Kantorovich potentials that is continuous at $\theta_0$. Since we assumed that for any $\theta \in V$, the Kantorovich potential for $W(\mu_\theta, \nu)$ is unique up to additive constants, there is a unique $\psi_{*\theta} \in \arg\max F(\cdot, \theta)$ such that $\psi_{*\theta}(y_1) = 0$, where $y_1$ is an arbitrary point in $\mathcal{Y}$. The continuity of $\theta \mapsto \psi_{*\theta}$ then follows from Lemma 8 given in Appendix C and the fact that $\theta \mapsto \mu_\theta$ is weak-$\star$ continuous at $\theta_0$. Indeed, Hypothesis 3 implies that when $\theta \to \theta_0$, $\mathbb{E}[\|g_\theta(Z) - g_{\theta_0}(Z)\|] \to 0$, which means that $g_\theta(Z) \to g_{\theta_0}(Z)$ in $L^1(\zeta, \mathbb{R}^d)$ and thus in distribution.

Lemma 3 then ensures that $\nabla_\theta F(\psi, \theta)$ exists in the neighborhood of $(\psi_{*0}, \theta_0)$.

Finally, we have to demonstrate that $(\psi, \theta) \mapsto \nabla_\theta F(\psi, \theta)$ is continuous at $(\psi_{*0}, \theta_0)$. For that, we show that the function $f(\psi, \theta, Z)$ introduced in Equation (37) has a simple expression in the neighborhood of $(\psi_{*0}, \theta_0)$. Indeed, for $\zeta$-almost all $z$, we can define $y = T_{\psi_{*0}}(g_{\theta_0}(z))$. Lemma 1 gives a neighborhood $\mathsf{W}_1 \times \mathsf{W}_2$ of $(\psi_{*0}, g_{\theta_0}(z))$ where $T_\psi(x) = y$. By continuity of $g(\cdot, z)$, $U = \mathsf{W}_1 \times (g(\cdot, z)^{-1}(\mathsf{W}_2))$ is a neighborhood of $(\psi_{*0}, \theta_0)$ such that $\forall (\psi, \theta) \in U$, $T_\psi(g_\theta(z)) = y$. Thus,

$$\forall (\psi, \theta) \in U, \quad \nabla_\theta f(\psi, \theta, z) = D_\theta g(\theta, z)^T \nabla \psi^c(g_\theta(z)) = D_\theta g(\theta, z)^T \nabla_x c(g_\theta(z), y). \tag{46}$$

In the neighborhood $U$, the right-hand side does not depend on $\psi$ anymore. It is also continuous in $\theta$ thanks to Hypothesis ($\mathsf{G}_\Theta$) and the fact that $c$ is $x$-regular. This proves that almost surely $\nabla_\theta f(\cdot, \cdot, Z)$ is continuous at $(\psi_{*0}, \theta_0)$. Finally, Equation (40) proves that all components of $\nabla_\theta f(\cdot, \cdot, Z)$ are almost surely bounded by $LK(Z) \in L^1(\zeta)$. Therefore, the dominated convergence theorem ensures that $\nabla_\theta F(\psi, \theta) = \mathbb{E}[\nabla_\theta f(\psi, \theta, Z)]$ is continuous at $(\psi_{*0}, \theta_0)$.

We can thus apply the envelope Theorem 7 which gives the desired result. □

The hypothesis that $c$ is $x$-regular (see Definition 1) may not be satisfied for some specific costs, like $c(x, y) = \|x - y\|$ in $\mathbb{R}^d$, which is related to the 1-Wasserstein distance. We now give a similar result with a relaxed hypothesis that encompasses such non-smooth cost functions.

**Theorem 4.** *Assume that $\mathcal{Y}$ is finite with $J$ points. Assume that $c : \mathcal{X} \times \mathcal{Y} \to \mathbb{R}$ is continuous, and that there is a constant $L > 0$, an open set $U$ with $\mathcal{X} \subset U \subset \mathbb{R}^d$ and a set $B \subset \mathcal{X}$ closed in $\mathbb{R}^d$ such that for any $y \in \mathcal{Y}$, $c(\cdot, y)$ can be extended to a $L$-Lipschitz $\mathscr{C}^1$ function on $U \setminus B$. Assume that $g$ satisfies the three assumptions of Theorem 3, and assume also that $\mu_\theta(B) = 0$ for any $\theta \in \Theta$.*

*Then $h(\theta) = W(\mu_\theta, \nu)$ is differentiable at any $\theta \in \Theta$ with the gradient expression (45).*

In the case of the Euclidean distance $c(x, y) = \|x - y\|$ in $\mathbb{R}^d$, for any $y \in \mathcal{Y}$, $c(\cdot, y)$ is $\mathscr{C}^1$ only on $\mathbb{R}^d \setminus \{y\}$. Thus this cost satisfies the condition of the last theorem with $B = \mathcal{Y}$, and the resulting hypothesis on the generator reads as $\mu_\theta(\mathcal{Y}) = 0$, in addition to the fact that the Laguerre interface $A_\psi$ is $\mu_\theta$-negligible.

*Proof.* One may check that the main parts of the proof given for Theorem 3 are still true with the relaxed hypothesis on the cost. We only highlight the minor modifications. First, in Lemma 2, one obtains that the $c$-transforms $\psi^c$ are only $\mathscr{C}^1$ on $\mathcal{X} \setminus (A_\psi \cup B)$. The proof of Lemma 3 is unchanged because $g_\theta(Z)$ almost surely belongs to $\mathcal{X} \setminus (A_\psi \cup B)$ where the $c$-transforms are differentiable. Finally, in the last step of the proof of Theorem 3, the neighborhood $\mathsf{W}_1 \times \mathsf{W}_2$ has to be replaced by $\mathsf{W}_1 \times \mathsf{W}_2 \cap B^c$. □

## 4 Gradient formula for regularized optimal transport

In this section, we show the gradient formula (Grad-OT) in the case of regularized optimal transport. With regularized OT, as soon as the cost is regular, the regularized $c$-transforms are also regular everywhere (see Lemma 6 in Appendix C). Hence no assumption are required on the target measure $\nu$, and the study is not restricted to the semi-discrete setting.

We recall the notations $W_\lambda$, $h_\lambda$, and $F_\lambda$ from (9), (16) and (19). In order to obtain the gradient of $h_\lambda$, we follow a similar strategy than in Section 3 and study the relation between $\nabla h_\lambda(\theta)$ and $\nabla_\theta F_\lambda(\psi, \theta)$. We first

show the gradient formula for the unregularized Wasserstein loss function in Section 4.1 and then extend this result for Sinkhorn divergences in Section 4.2.

## 4.1 Gradient of the loss with entropic regularization

We first recall a result from Genevay et al. (2019) on the gradient of regularized $c$-transforms and detail its proof for further use.

**Lemma 4** (Gradient of regularized $c$-transforms Genevay et al. (2019))**.** *Let $\lambda > 0$. Assume that $c$ is $x$-regular (see Definition 1). Let $\psi \in \mathscr{C}(\mathcal{Y})$. Then $\psi^{c,\lambda}$ is $\mathscr{C}^1$ on $\mathcal{X}$ and*

$$\forall x \in \mathcal{X}, \quad \nabla \psi^{c,\lambda}(x) = \int_{\mathcal{Y}} \exp\left(\frac{\psi^{c,\lambda}(x) + \psi(y) - c(x,y)}{\lambda}\right) \nabla_x c(x,y) d\nu(y) \tag{47}$$

$$= \frac{\int_{\mathcal{Y}} \exp\left(\frac{\psi(y)-c(x,y)}{\lambda}\right) \nabla_x c(x,y) d\nu(y)}{\int_{\mathcal{Y}} \exp\left(\frac{\psi(y)-c(x,y)}{\lambda}\right) d\nu(y)}. \tag{48}$$

*Proof.* By definition of the regularized $c$-transform (see relations (12) and (13)), we have for any $x \in \mathcal{X}$,

$$\forall x \in \mathcal{X}, \quad e^{-\frac{\psi^{c,\lambda}(x)}{\lambda}} = \int_{\mathcal{Y}} e^{\frac{\psi(y)-c(x,y)}{\lambda}} d\nu(y). \tag{49}$$

Since $c$ is $x$-regular, then for any $y \in \mathcal{Y}$, $x \mapsto e^{\frac{\psi(y)-c(x,y)}{\lambda}}$ is $\mathscr{C}^1$ on a neighborhood of $\mathcal{X}$, and it is Lipschitz with Lipschitz constant $\frac{L}{\lambda} \exp(\frac{\|\psi\|_\infty + \|c\|_\infty}{\lambda})$. Therefore, we can differentiate under the integral to get

$$\forall x \in \mathcal{X}, \quad \nabla_x \left(e^{-\frac{\psi^{c,\lambda}(x)}{\lambda}}\right) = -\frac{1}{\lambda} \int_{\mathcal{Y}} \nabla_x c(x,y) e^{\frac{\psi(y)-c(x,y)}{\lambda}} d\nu(y). \tag{50}$$

Expanding the left-hand side, this is equivalent to

$$\forall x \in \mathcal{X}, \quad \nabla \psi^{c,\lambda}(x) e^{-\frac{\psi^{c,\lambda}(x)}{\lambda}} = \int_{\mathcal{Y}} \nabla_x c(x,y) e^{\frac{\psi(y)-c(x,y)}{\lambda}} d\nu(y), \tag{51}$$

which gives the desired formula (47) for $\nabla \psi^{c,\lambda}$. The second expression (48) follows from using the definition of $\psi^{c,\lambda}$ again. Finally, using (48), one can see that all integrated functions are continuous with respect to $x$, and bounded. The dominated convergence theorem thus ensures that $\nabla \psi^{c,\lambda}$ is continuous. $\qquad\square$

We now look at the differentiation of $F_\lambda$ before showing the gradient formula (Grad-OT).

**Lemma 5** (Regularity of $F_\lambda$)**.** *Let $\lambda > 0$. Assume that $c$ is $x$-regular and that $g$ satisfies Hypothesis ($\mathsf{G}_\Theta$) (Definition 3). Let $\psi \in \mathscr{C}(\mathcal{Y})$. Then the function $\theta \mapsto F_\lambda(\psi, \theta)$ is differentiable on $\Theta$ and*

$$\forall \theta_0 \in \Theta, \quad \nabla_\theta F_\lambda(\psi, \theta_0) = \mathbb{E}\left[D_\theta g(\theta_0, Z)^T \nabla \psi^{c,\lambda}(g(\theta_0, Z))\right]. \tag{52}$$

*Proof.* As in the proof of Lemma 3, let us introduce

$$f_\lambda(\psi, \theta, Z) = \psi^{c,\lambda}(g_\theta(Z)) \tag{53}$$

so that $F_\lambda(\psi, \theta) = \mathbb{E}[f_\lambda(\psi, \theta, Z)]$. Using Hypothesis ($\mathsf{G}_\Theta$) and Lemma 4, $f_\lambda(\psi, \cdot, Z)$ is differentiable on $\Theta$ almost surely, and, thanks to the chain-rule,

$$\nabla_\theta f_\lambda(\psi, \theta, Z) = D_\theta g(\theta, Z)^T \nabla \psi^{c,\lambda}(g_\theta(Z)). \tag{54}$$

Besides, the regularized $c$-transforms of a $x$-regular cost being still $L$-Lipschitz (by Lemma 6 in Appendix C), we have an integrable bound for the finite differences of $f_\lambda$. Indeed, for a fixed $\theta_0 \in \Theta$, and for any $\theta$ in the neighborhood $V$ of $\theta_0$ given by Hypothesis ($\mathsf{G}_{\theta_0}$), we have

$$|f_\lambda(\psi, \theta, Z) - f_\lambda(\psi, \theta_0, Z)| \leqslant L\|g(\theta, Z) - g(\theta_0, Z)\| \leqslant LK(Z)\|\theta - \theta_0\| \quad \text{a.s.} \tag{55}$$

The proof can then be ended as the one of Lemma 3. For further use, notice that the previous bound implies

$$\forall \theta_0 \in \Theta, \quad \|\!\|\nabla_\theta f_\lambda(\psi, \theta_0, Z)\|\!\| = \|\!\|D_\theta g(\theta_0, Z)^T \nabla \psi^{c,\lambda}(g(\theta_0, Z))\|\!\| \leqslant LK(Z) \quad \text{a.s.} \tag{56}$$

where $\|\!\|\cdot\|\!\|$ is the dual norm associated with $\|\cdot\|$. $\qquad\square$

**Theorem 5.** *Assume that $c$ is $x$-regular and that $g$ satisfies* Hypothesis $(\mathsf{G}_\Theta)$. *Then $h_\lambda$ is $\mathscr{C}^1$ on $\Theta$ and*

$$\forall \theta \in \Theta, \quad \nabla_\theta h_\lambda(\theta) = \nabla_\theta F_\lambda(\psi_*, \theta) = \mathbb{E}\left[D_\theta g(\theta, Z)^T \nabla \psi_*^{c,\lambda}(g(\theta, Z))\right], \tag{57}$$

*where $\psi_* \in \arg\max_\psi H_\lambda(\psi, \theta)$, and where $\nabla \psi_*^{c,\lambda}$ is given by (47).*

*Proof.* As in Theorem 3, the proof is based on the envelope Theorem 7, given in Appendix D, applied on $h_\lambda(\theta) = W_\lambda(\mu_\theta, \nu) = \max_\psi H_\lambda(\psi, \theta)$ (see (18) and (20)). With entropic regularization, the uniqueness (up to additive constants) of the solutions of the dual problem provides a continuous selection of Kantorovich potentials, thanks to Lemma 8 in Appendix C. Besides, Lemma 5 ensures that $\nabla_\theta F_\lambda(\psi, \theta)$ exists for any $\psi \in \mathscr{C}(\mathcal{Y})$ and any $\theta \in \Theta$. It remains to show that $\nabla_\theta F_\lambda$ is continuous in $(\psi, \theta)$. For that, recall that $\nabla_\theta F_\lambda(\psi, \theta) = \mathbb{E}[\nabla_\theta f_\lambda(\psi, \theta, Z)]$ where $f_\lambda$ is defined in (53), and let us fix $\theta_0 \in \Theta$ and $\psi_{*0}$ an optimal Kantorovich potential for $W_\lambda(\mu_{\theta_0}, \nu)$, and let also $\psi \in \mathscr{C}(\mathcal{Y})$ be arbitrary. Then, for any $\theta$ in the neighborhood $V$ of $\theta_0$ given by Hypothesis $(\mathsf{G}_{\theta_0})$,

$$|f_\lambda(\psi, \theta, Z) - f_\lambda(\psi, \theta_0, Z)| \leqslant L\|g(\theta, Z) - g(\theta_0, Z)\| \leqslant LK(Z)\|\theta - \theta_0\| \quad \text{a.s.} \tag{58}$$

which ensures that all components of $\nabla_\theta f_\lambda(\psi, \theta, Z)$ are almost surely bounded by $LK(Z)$, an integrable bound which does not depend on $(\psi, \theta)$. Since $\theta \mapsto g(\theta, Z)$ is almost surely $\mathscr{C}^1$ on $V$, using (54), the last step is to show that $(\psi, \theta) \mapsto \nabla \psi^{c,\lambda}(g_\theta(Z))$ is almost surely continuous.

Let us fix $z \in \mathcal{Z}$ for which $g(\cdot, z)$ is $\mathscr{C}^1$ on $V$ and for which *Hypothesis $(\mathsf{G}_\Theta)$* (21) holds. From (51), we have

$$\nabla \psi^{c,\lambda}(g_\theta(z)) = e^{\frac{\psi^{c,\lambda}(g_\theta(z))}{\lambda}} \int_{\mathcal{Y}} \nabla_x c(g_\theta(z), y) e^{\frac{\psi(y) - c(g_\theta(z), y)}{\lambda}} \, d\nu(y). \tag{59}$$

For the first term, if $\psi, \chi \in \mathscr{C}(\mathcal{Y})$ and $\theta, \tau \in \Theta$,

$$|\psi^{c,\lambda}(g_\theta(z)) - \chi^{c,\lambda}(g_\tau(z))| \leqslant |\psi^{c,\lambda}(g_\theta(z)) - \psi^{c,\lambda}(g_\tau(z))| + |\psi^{c,\lambda}(g_\tau(z)) - \chi^{c,\lambda}(g_\tau(z))| \tag{60}$$

$$\leqslant L|g_\theta(z) - g_\tau(z)| + \|\psi^{c,\lambda} - \chi^{c,\lambda}\|_\infty \tag{61}$$

$$\leqslant LK(z)\|\theta - \tau\| + \|\psi - \chi\|_\infty , \tag{62}$$

by virtue of Lemma 7 given in Appendix C and thus $(\psi, \theta) \mapsto e^{\frac{\psi^{c,\lambda}(g_\theta(z))}{\lambda}}$ is continuous. For the integral term, $\theta \mapsto \nabla_x c(g_\theta(z), y)$ is continuous on $V$ because $c$ is $x$-regular and because $\theta \mapsto g(\theta, z)$ is continuous on $V$. Additionally, $(\psi, \theta) \mapsto e^{\frac{\psi(y) - c(g_\theta(z), y)}{\lambda}}$ is a separable product of two terms which are continuous in $\psi$ and $\theta$ respectively. Finally, if $\psi$ is restricted in a neighborhood $P_0$ of $\psi_{*0}$ bounded by $R > 0$, then all terms under the integral are bounded by $Le^{\frac{1}{\lambda}(R + \|c\|_\infty)}$. Again, the dominated convergence theorem implies that $(\psi, \theta) \mapsto \nabla \psi^{c,\lambda}(g_\theta(z))$ is continuous on $P_0 \times V$. Finally, using the bound (56) and the dominated convergence theorem gives that $(\psi, \theta) \mapsto \nabla_\theta F_\lambda(\psi, \theta)$ is continuous on $P_0 \times V$. Therefore, all required conditions are satisfied to apply the envelope theorem on $F_\lambda$, which gives the desired formula. $\qquad\square$

## 4.2 Gradient of the Sinkhorn divergence

We here assume that $\mathcal{X} = \mathcal{Y}$, that the cost $(x, y) \mapsto c(x, y)$ is $\mathscr{C}^1$ on $\mathcal{X} \times \mathcal{X}$ and symmetric (*i.e.* $c(x, y) = c(y, x)$ for all $x, y \in \mathcal{X}$) and that it is $L$-Lipschitz with respect to $(x, y)$:

$$\forall (x, y), (x', y') \in \mathcal{X} \times \mathcal{X}, \quad |c(x, y) - c(x', y')| \leqslant L(\|x - x'\| + \|y - y'\|). \tag{63}$$

Genevay et al. (2018) and Feydy et al. (2019) have shown that learning a generative model based on the Wasserstein cost $W_\lambda$ induces a bias. For this reason, they propose to use instead the so-called Sinkhorn divergence defined as

$$S_\lambda(\mu, \nu) = W_\lambda(\mu, \nu) - \frac{1}{2}\Big(W_\lambda(\mu, \mu) + W_\lambda(\nu, \nu)\Big). \tag{64}$$

Since we focus here on the regularized WGAN learning problem, we study the function

$$s_\lambda(\theta) = S_\lambda(\mu_\theta, \nu) = W_\lambda(\mu_\theta, \nu) - \frac{1}{2}\Big(W_\lambda(\mu_\theta, \mu_\theta) + W_\lambda(\nu, \nu)\Big) \tag{65}$$

and we extend the previous regularity results to this new criterion. The first term of (65) has already been studied, and the last term does not depend on $\theta$. It thus remains to study the middle term. To this end, we rely on the dual formulation of $W_\lambda(\mu_\theta, \mu_\theta)$ given by

$$W_\lambda(\mu_\theta, \mu_\theta) = \max_{\chi, \eta \in \mathscr{C}(\mathcal{X})} \mathcal{F}_\lambda(\chi, \eta, \theta) := \int_\mathcal{X} \chi d\mu_\theta + \int_\mathcal{X} \eta d\mu_\theta - \lambda \int_{\mathcal{X} \times \mathcal{X}} e^{\frac{\chi(x)+\eta(y)-c(x,y)}{\lambda}} d\mu_\theta(x) d\mu_\theta(y) + \lambda. \tag{66}$$

The problem (66) can be restricted to functions which are regularized $c$-transforms with respect to $\mu_\theta$. Similar to Lemma 5, one can show that such regularized $c$-transforms are still $\mathscr{C}^1$ on $\mathcal{X}$ and that the gradient can be computed by differentiating under the integral. Besides, because of the symmetry of $W_\lambda(\mu_\theta, \mu_\theta)$, one can show that the couple of optimal potentials $(\chi_*, \eta_*)$ that solve (66) is such that $\chi_*$ and $\eta_*$ are equal up to an additive constant. Based on these observations, we can now state the regularity result for the Sinkhorn divergence. Recall that notations $I$, $F_\lambda$ are defined in (17) and (19).

**Theorem 6.** *Assume that $c$ is $x$-regular and that $g$ satisfies Hypothesis ($\mathsf{G}_\Theta$). Then $s_\lambda$ is $\mathscr{C}^1$ on $\Theta$ and*

$$\forall \theta \in \Theta, \quad \nabla_\theta s_\lambda(\theta) = \nabla_\theta F_\lambda(\psi_*, \theta) - \nabla_\theta I(\chi_*, \theta) = \mathbb{E}\left[D_\theta g(\theta, Z)^T \left(\nabla \psi_*^{c,\lambda}(g(\theta, Z)) - \nabla \chi_*(g(\theta, Z)))\right)\right], \tag{67}$$

*where $\psi_* \in \arg\max_\psi F_\lambda(\psi, \theta)$ and $(\chi_*, \eta_*) \in \arg\max_{(\chi,\eta)} \mathcal{F}_\lambda(\chi, \eta, \theta)$.*

The proof is postponed to Appendix E.

## 5    Experiments

The goal of this section is to provide a practical analysis of the proposed approach for learning a generative model by means of the regularized optimal transport framework. More precisely, we aim at examining the impact of the regularization parameter and the choice of parameterization for the dual variable of the corresponding optimization problem. We first provide in Section 5.1 an alternating algorithm that tackles the minimization of (16) for learning a generator from a dataset. In Section 5.2, we apply this algorithm to synthetic datasets with a few number of points in dimension 2. This helps to understand and explain the possible trajectories of the alternating algorithm. Then we consider in Section 5.3 the MNIST dataset (LeCun et al., 1998) composed of 60000 digits in dimension 784.

### 5.1    Alternating Algorithm

In the semi-discrete setting, $\mathcal{Y}$ is a finite set of $J$ data points, and elements $\psi \in \mathscr{C}(\mathcal{Y})$ identifies to vectors in $\mathbb{R}^J$ as in Section 3. On the other hand, $\mu_\theta$ is the output distribution of a generative network, which can be sampled on demand. We thus aim at solving the optimization problem

$$\min_{\theta \in \Theta} h_\lambda(\theta) = \min_{\theta \in \Theta} \max_{\psi \in \mathscr{C}(\mathcal{Y})} H_\lambda(\psi, \theta) := \int_\mathcal{X} \psi^{c,\lambda} d\mu_\theta + \int_\mathcal{Y} \psi d\nu = F_\lambda(\psi, \theta) + \sum_{y \in \mathcal{Y}} \psi(y)\nu(\{y\}) . \tag{68}$$

We adopt an algorithm that alternates between updating $\theta$ with one gradient step, and updating $\psi$ with several iterations of a dedicated algorithm.

As shown in our main theorems (see relations (45) and (57)), computing the gradient of $h_\lambda$ requires an optimal dual potential $\varphi_* = \psi_*^{c,\lambda}$. In general, we only have access to an approximation of $\varphi_*$. In our work, we rely on the stochastic algorithm for semi-discrete OT proposed in (Genevay et al., 2016) to approximate the optimal potential. We recall that from the notation in the proof of Lemma 5, we have

$$F_\lambda(\psi, \theta) = \mathbb{E}[f_\lambda(\psi, \theta, Z)] \tag{69}$$

where $f_\lambda(\psi, \theta, z) = \psi^{c,\lambda}(g_\theta(z))$. It has been shown in (Genevay et al., 2016; Houdard et al., 2023) that $H_\lambda(\cdot, \theta)$ defined in (68) is a concave function whose supergradient $\mathcal{D}(\psi, \theta) = \partial_\psi H_\lambda(\psi, \theta)$ can be written as

$$\mathcal{D}(\psi, \theta) = \mathbb{E}[\mathsf{D}(\psi, \theta, Z)] \quad \text{where} \quad \mathsf{D}(\psi, \theta, z) = \partial_\psi \left( f_\lambda(\psi, \theta, z) - \int \psi d\nu \right). \tag{70}$$

This element $\mathsf{D}(\psi, \theta, z) \in \mathbb{R}^J$ can be computed with an explicit formula given in (Genevay et al., 2016; Houdard et al., 2023). It is in practice implemented by automatic differentiation.

Therefore, for a current $\theta$, we can optimize $\psi$ with a stochastic supergradient ascent (for which we adopt the common referencing ASGD for averaged stochastic gradient descent)

$$\begin{cases} \widetilde{\psi}_k & = \widetilde{\psi}_{k-1} + \frac{\gamma}{k^\alpha} \left( \frac{1}{|B_k|} \sum_{z \in B_k} \mathsf{D}(\widetilde{\psi}_{k-1}, \theta, z) \right) \\ \psi_k & = \frac{1}{k}(\widetilde{\psi}_1 + \cdots + \widetilde{\psi}_k), \end{cases} \tag{71}$$

where $\gamma > 0$ is the learning rate, $\alpha \in (0, 1)$, $B_k$ is a batch containing $b$ independent samples of the distribution of $Z$, and the different batches $B_k$'s are also independent. In ASGD, the variable $\widetilde{\psi}_0$ can be initialized at 0 (cold start), or alternately at the value of $\psi$ obtained by ASGD for the previous $\theta$ (warm start). As recalled in (Genevay et al., 2016) and (Galerne et al., 2018), for $\alpha = 0.5$, the ASGD algorithm has a convergence guarantee in $\mathcal{O}(\frac{\log k}{\sqrt{k}})$ on the function values. After $K$ iterations of the inner loop, we obtain an approximation $\underline{\psi}$ of the optimal dual potential $\psi_*$ for $W_\lambda(\mu_\theta, \nu)$, and we use it to perform the gradient descent step on $\theta$

$$\nabla_\theta h_\lambda(\theta) \approx \nabla_\theta H_\lambda(\underline{\psi}, \theta) = \nabla_\theta F_\lambda(\underline{\psi}, \theta) = \mathbb{E}\left[ D_\theta g(\theta, Z)^T \nabla \underline{\psi}^{c,\lambda}(g(\theta, Z)) \right] \tag{72}$$

Using $\nabla_\theta H_\lambda(\underline{\psi}, \theta)$ as a proxy for $\nabla_\theta h_\lambda(\theta)$ in the gradient descent step can be interpreted as an alternating optimization on $H_\lambda(\psi, \theta)$, with an inner loop on $\psi$ at each iteration. As the expectation in (72) cannot be computed in closed form, we realize an approximation by taking another batch $B'$ on $z$ (drawn with distribution $\zeta$):

$$\nabla_\theta H_\lambda(\underline{\psi}, \theta) \approx \frac{1}{|B'|} \sum_{z \in B'} D_\theta g(\theta, z)^T \nabla \underline{\psi}^{c,\lambda}(g(\theta, z)). \tag{73}$$

The overall method, summarized in Algorithm 1, can be interpreted as a stochastic alternating optimization algorithm on a fixed cost $H_\lambda(\psi, \theta)$. This generalizes the related work of (Chen et al., 2019) which is restricted to unregularized OT $\lambda = 0$. The stochastic gradient steps taken on $\psi$ and $\theta$ can be implemented by automatic differentiation on a fixed cost $H_\lambda$, and the updates can be implemented with predefined optimizers. A neural network parameterization $\psi_t$ of the variable $\psi$ can also be adopted as in (Seguy et al., 2018). The update of $\psi$ then translates on an update of the neural network parameters $t$, which can be done by backpropagating the gradient of $t \mapsto f_\lambda(\psi_t, \theta, z) - \int \psi_t d\nu = \psi_t^{c,\lambda}(g_\theta(z)) - \sum_{y \in \mathcal{Y}} \psi_t(y)\nu(\{y\})$ through the differentiation of $f_\lambda(\psi, \theta, z) = \psi^{c,\lambda}(g_\theta(z))$ for a batch of $z$ values. The exact computation of $\psi^{c,\lambda}$ requires to visit all the dataset $\mathcal{Y}$, which is prohibitive for large databases. An alternative strategy is to work with a batchwise version of the OT cost, but this introduces an estimation bias Fatras et al. (2020).

---

**Algorithm 1** Alternating algorithm to learn generative model $\mu_\theta$

---

**Initialization:** $\psi_0 = 0$ and $\theta$ (randomly)

**For** $n = 1$ **to** $N$

- Approximate $\psi_n \approx \arg\max H_\lambda(\cdot, \theta)$: inner loop with $K$ iterations of ASGD using supergradient (70) on batches $B_{n,1}, \ldots, B_{n,K}$ of size $b$ on $z$ , initialized at $\psi_{n-1}$ (warm start) or at 0 (cold start)
- Update $\theta$ with one step of ADAM algorithm on $H_\lambda(\psi_n, \cdot)$ using gradient (73) on a batch $B'_n$ of size $b$ on $z$

**Output:** estimated generative model parameters $\theta$

---

## 5.2 Generator learning for Synthetic datasets

We examine the behavior of the alternating Algorithm 1 for fitting a generative model to a small 2D synthetic dataset ($J = 2$ points in Section 5.2.1 and $J = 6$ points in Section 5.2.2). For each experiment, we give

diagrams to illustrate the position of the dataset $\mathcal{Y}$ and the evolution of the support of $\mu_\theta$ along the iterations $n$ of the main loop of the algorithm. The square support of these diagrams corresponds to $[0,1]^2$. The Laguerre diagram associated with a given $\psi \in \mathbb{R}^J$ is drawn as a colored partition of $[0,1]^2$. On the right column of the figures, we display the evolution of the loss $H_\lambda(\psi, \theta)$. Since $H_\lambda(\psi, \theta)$ involves a $\mu_\theta$-integral, for plotting we always use a Monte-Carlo estimate obtained with a batch drawn from $\mu_\theta$.

We use the quadratic cost $c(x,y) = \|x - y\|^2$ and the settings detailed in Appendix F.1. Note that in Algorithm 1 the ADAM algorithm is used to update the model, which is often parametrized by a neural network, as done in the following with an MLP generator. However, when considering simpler models (dirac, disc and ellipse distributions that are explicitly parametrized), we rely on a stochastic gradient descent (SGD) algorithm with fixed learning rate $\eta = 0.2$. In both cases, the dual variable $\psi$ is optimized with the ASGD algorithm written in (71) with learning rate $\gamma$ and a cold start initialization. The number $K$ of ASGD iterations is always fixed to $K = 100$.

### 5.2.1 2-point dataset

We use the generative models detailed in Appendix F.1 for a target distribution $\nu$ uniform on $\mathcal{Y} = \{y_1, y_2\}$.

**Dirac generator**   We first study the example of Proposition 2 and consider the Dirac generator $\mu_\theta = \delta_\theta$. In this case, the gradient descent on $\theta$ is actually deterministic. Besides, from the calculations of Section 2.3, we have that the global minimum of $h_\lambda(\theta) = W_\lambda(\mu_\theta, \nu)$ is reached at $\theta_* = \frac{y_1 + y_2}{2}$.

Fig. 3 displays the results for the Dirac generator. It illustrates the position of the estimated Laguerre diagram along iterations. Let us recall that, as illustrated in Fig. 1 for a fixed $\theta$, the solution of the dual transportation problem makes the Laguerre interface go through $\theta$. This is not the case when using Algorithm 1, even in this deterministic setting. Hence the current point $\theta$ can get close to the Laguerre interface but it does not get caught by this interface. Indeed, with the chosen initialization of $\theta$ and gradient strategies, the combination of gradient steps on $\psi$ and $\theta$ has no reason to put the interface exactly on $\theta$.

In view of Proposition 2, it means that for a current $\theta$, the ASGD on $\psi$ does not reach the optimum $\psi_*$ in finite time. Hence the corresponding interface does not pass through $\theta$ and the algorithm does not get trapped in a point where the gradient of $\theta \mapsto H_0(\psi, \theta)$ does not exist. This illustrates that it is very unlikely to encounter the gradient problem highlighted in Proposition 2 in real use cases. In contrast, the gradient problem appears when taking an optimal gradient step on $\psi$ instead of a fixed step size. Indeed, with such a strategy, one iteration of $\psi$ puts the interface exactly on $\theta$, which prevents from computing the next gradient.

Fig. 3 exhibits four different behaviors of the alternating algorithm: without regularization ($\lambda = 0$) in the three first rows, and with $\lambda = 0.1$ in the last one. In the first row, the alternating algorithm stabilizes, but for a wrong reason: the learning rate $\gamma$ of ASGD is too small, and thus the interface will not get past $\theta$. Therefore, $\theta$ being always in the same Laguerre cell, the gradient descent on $\theta$ makes the generator collapse to the corresponding data point $y_j$, which is not even a local minimum of $h_\lambda$.

An opposite phenomenon appears in the second row, where $\gamma$ is larger: the algorithm now oscillates. In this case, because of the fixed number of iterations of the inner loop, ASGD always makes a non negligible error. This reflects in the position of the interface, which hardly get close to $\theta$, and actually moves periodically on opposite sides of $\theta$. Thus, the point $\theta$ is alternately pushed towards $y_1$ and $y_2$, which corresponds to the oscillations in the evolution of the loss function. This example shows that, **with a fixed step size strategy, the alternating algorithm may not converge in certain cases**. Note that using a decreasing step, which is often used to mitigate oscillating behaviour during stochastic optimization, does not necessarily solve this issue. As illustrated in the third row, with a carefully tuned gradient step strategy and smaller learning rates $\eta$, the algorithm stabilizes around the optimal position $\theta_*$, with slight oscillations.

Finally, for the regularized case shown in the fourth row, one can see that $\theta$ quickly gets close to the global optimum $\theta_* = \frac{y_1 + y_2}{2}$. Indeed, from the gradient formula (57), the $\nabla_\theta$-descent direction points to a weighted average of $y_1 - \theta$ and $y_2 - \theta$ at each iteration, which helps to push $\theta$ towards $\theta_*$. One can observe that using the transported measure $T_\psi \sharp \mu_\theta$ never solves the WGAN problem because it would just copy $y_1$ or $y_2$ whereas the true solution $\delta_{\theta_*}$ realizes a compromise between all data points. The proposed alternating algorithm is

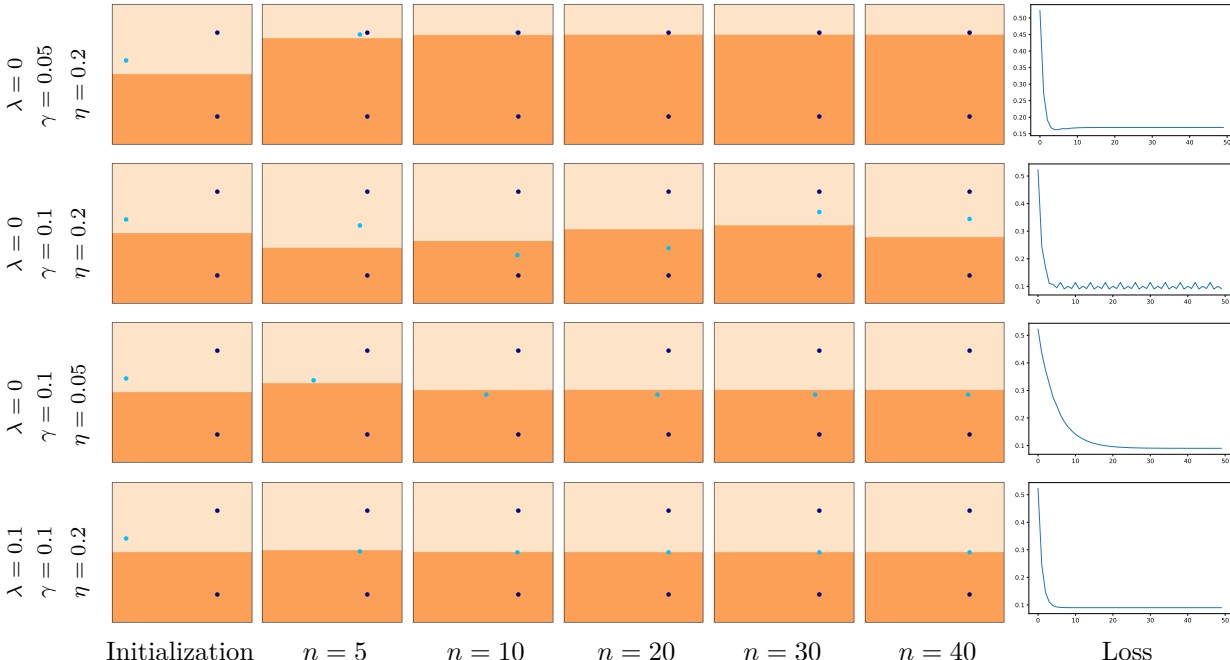

Figure 3: **Generator learning for** 2**-point dataset** $\mathcal{Y}$ **and Dirac generator** $\delta_\theta$. In the first columns, we display $\mathcal{Y}$ (dark blue) and the current position of $\theta$ (light blue) along iterations $n$. The Laguerre diagram associated to the current value of $\underline{\psi}_\theta$ is displayed in the background with the colored partition. The last column shows the evolution of the loss $H_\lambda(\underline{\psi}_\theta, \theta)$ along the iterates. For each line, the regularization parameter $\lambda$ and the learning rates $\gamma, \eta$ for $\psi$ and $\theta$ are indicated on the left. Depending on the choice of parameters and gradient strategy, the alternating algorithm can oscillate or stabilize. See the text for additional comments.

a relevant way to find this compromise, provided that its hyperparameters are carefully tuned. Besides, this example illustrates that entropic regularization may help to reach this compromise faster, while introducing a non negligible bias in the problem as explained in Genevay et al. (2018).

**Disk generator**   We now consider the same data set $\mathcal{Y} = \{y_1, y_2\}$ and a generator $g_\theta$ inducing the uniform distribution $\mu_\theta = \mathcal{U}(D(\theta, r))$ on the disk of center $\theta$ and of fixed radius $r$. In this case, $\mu_\theta$ is absolutely continuous w.r.t. the Lebesgue measure; thus the SGD on $\theta$ is truly stochastic. All hypotheses of Theorem 3 are satisfied in this setting, which validates the gradient formula (Grad-OT) for all values of $\psi$ and $\theta$.

Different behaviors can nevertheless be observed with the alternating algorithm, as illustrated in Fig. 4 which present experiments conducted with $\lambda = 0$. If $r$ is sufficiently small (rows 1 and 2), the algorithm behaves as in the case of the Dirac generator: for large value of $\gamma$ the algorithm can oscillate around the optimal position $\frac{y_1 + y_2}{2}$ (row 1), and for low values of $\gamma$, it can make $\theta$ collapse to one of the data point (row 2). In this latter case, the "cold start" initialization $\psi = 0$ actually explains the bad positioning of the interface. With a "warm start" initialization of $\psi$ i (row 3), the estimated interface is indeed able to closely follow the position of $\theta$. However, this amplifies the oscillations of $\theta$, since the interface goes past $\theta$ more often.

Note that the oscillatory behavior is reduced when the radius $r$ of the disk is increased (rows 4 and 5). Indeed, if the disk is larger, the interface stays close to $D(\theta, r)$ as soon as they intersect. This illustrates that a largely spread distribution $\mu_\theta$ make the alternating algorithm stabilize faster, even when starting from a purely random initialization (row 5).

**Other generators**   We consider in Fig. 5 one ellipse generator (1-ellipse, row 1), one "mixture of two ellipses" generator (2-ellipse, rows 2 & 3), and a multilayer perceptron (MLP, rows 4 & 5). In these experiments, the generated distribution tends to concentrate either on $\mathcal{Y}$ or on a one-dimensional structure close

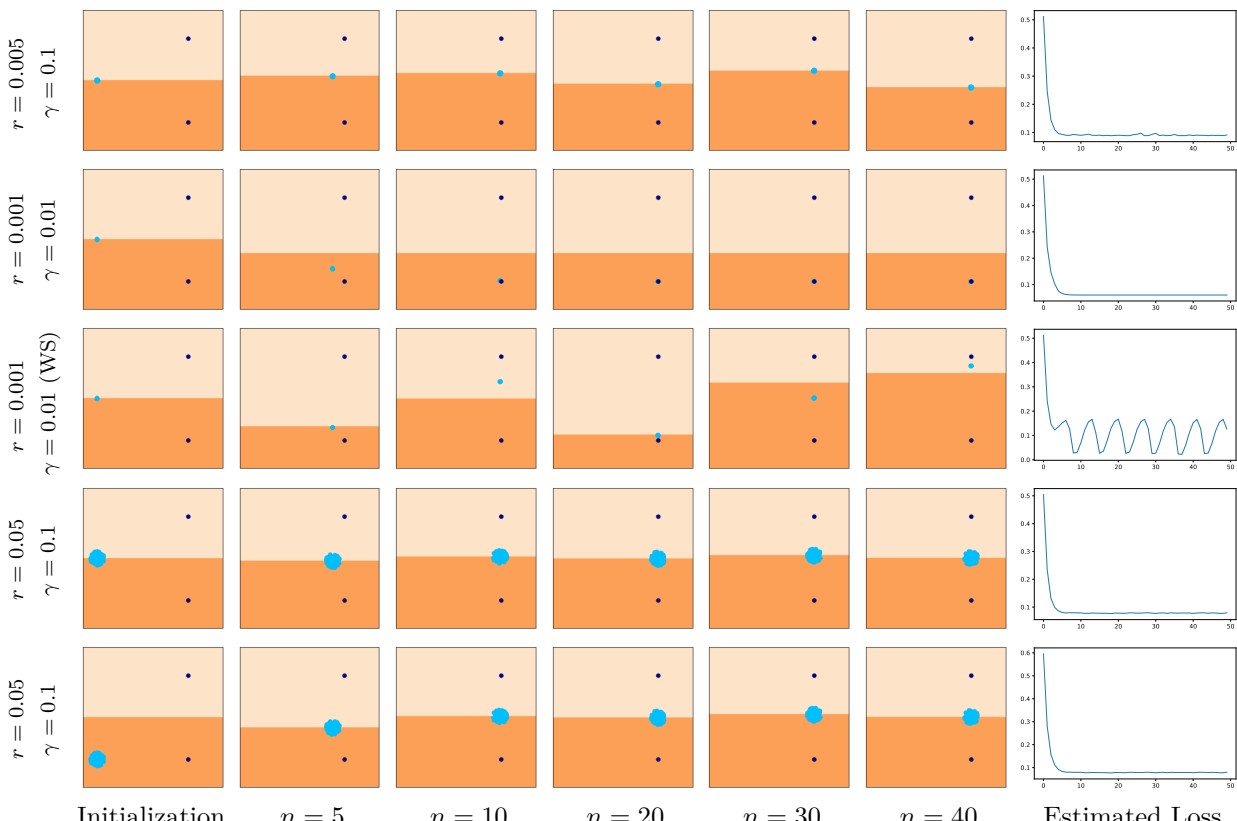

Figure 4: **Generator learning for $2$-point dataset $\mathcal{Y}$ and Disk generator centered at $\theta$.** Results are displayed using the same presentation as in Fig. 3, with samples from $\mu_\theta$ in light blue. For each line, the disk radius $r$ and the learning rate $\gamma$ are indicated on the left. The regularization parameter is here fixed to $\lambda = 0$. The last column shows the evolution of the estimated loss $H_\lambda(\underline{\psi}_\theta, \theta)$ along the iterates. In the third line, we used a "warm start" (WS) initialization for $\psi$ at each iteration, as opposed to "cold start" in other cases (see Alg. 1). Depending on the choice of parameters and gradient strategy, the alternating algorithm may oscillate or stabilize (possibly to a sub-optimal configuration). See the text for additional comments.

to the line $(y_1, y_2)$. The difference between first rows is worth noticing: if possible, the generator collapses on the data points in order to decrease the $W^2$ cost. This is possible in the 2-ellipse case (rows 2, 3), but not in the 1-ellipse case (row 1). This illustrates that the choice of the generative model is important and that mode collapse cannot be only attributed to the loss function, *i.e.* the $W^2$ cost or its regularized version.

### 5.2.2  $6$-**point dataset, neural generator**

We now consider a dataset $\mathcal{Y}$ with 6 points, and we fit a generative model parameterized by a multilayer perceptron (MLP), whose parameters can be found in Appendix F.1. The results are displayed in Fig. 6 below, and in Fig. 11, Fig. 12 of Appendix G. On Fig. 6, the estimated generative model concentrates on a shape, which is close to the data points for $\lambda = 0$, while it degenerates to a barycenter for larger $\lambda$. This phenomenon is confirmed by the results of Fig. 11, and agrees with the fact that entropic regularization introduces a non negligible bias in the WGAN fitting problem, as already highlighted by Genevay et al. (2018). Correcting this bias requires the use of the Sinkhorn divergence $S_\lambda$ instead of $W_\lambda$. However, since the source measure $\mu_\theta$ is absolutely continuous, estimating $S_\lambda$ without relying on a batch approximation is not straightforward. For this reason, we do not address WGAN estimation with the Sinkhorn divergence.

In these experiments, the loss tends to oscillate. Following Section 5.2.1, this behavior can be explained by the large number of parameters of the MLP associated to the fixed number of iterations $K$ of the inner loop.

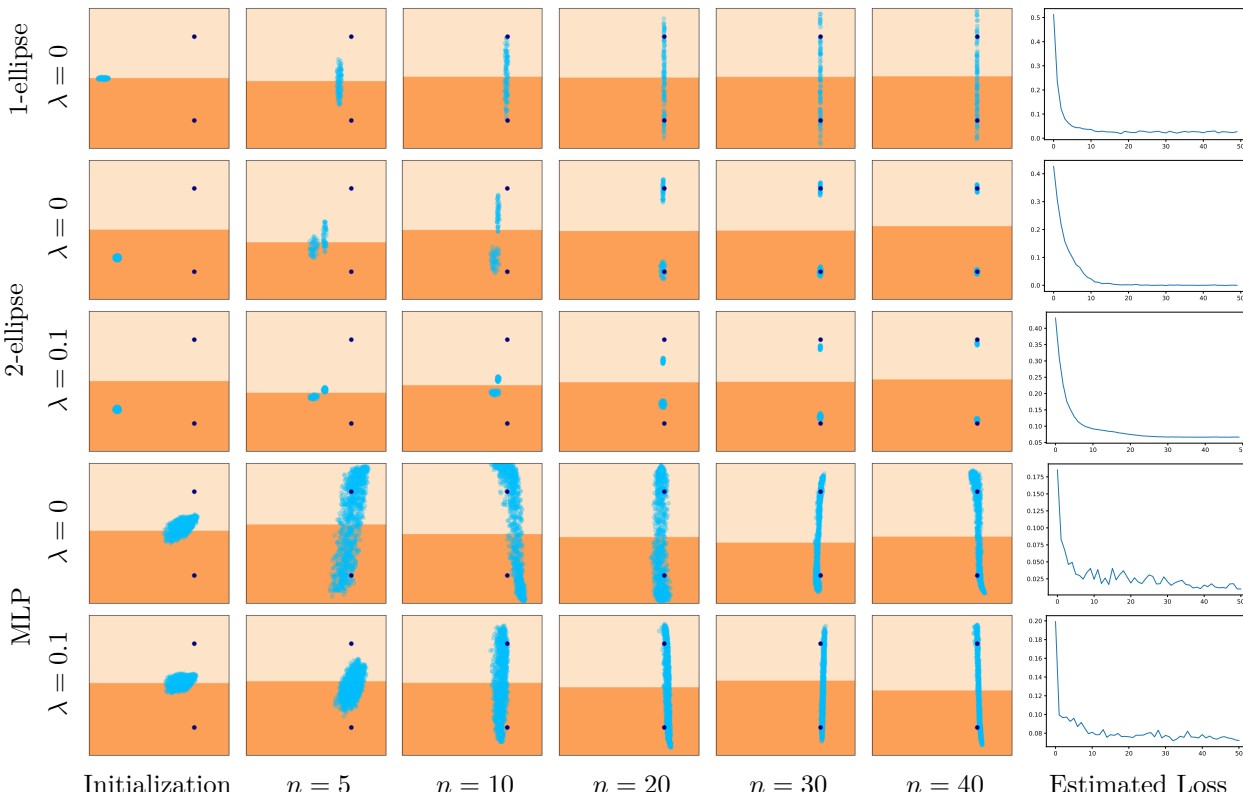

Figure 5: **Generator learning for 2-point dataset $\mathcal{Y}$ and various generators**: 1-ellipse (row 1), 2-ellipse (rows 2 & 3), and MLP (rows 4 & 5). Results are displayed using the same presentation as in Figure 4. The learning rate for $\psi$ is here fixed to $\gamma = 0.1$. See the text comments for more details.

In the first row of Fig. 6, the loss slightly oscillates for $K = 100$, end the generated distribution $\mu_\theta$ realizes a good approximation of $\nu$ for $n = 200$. In the second row, the number of iterations of the inner loop is reduced to $K = 2$ and the algorithm stabilizes to a bad configuration. This suggests that when $K$ is too low, the ASGD does not converge fast enough to irrigate all points of the target distribution $\nu$. If the Laguerre cell of one data point $y_j$ stays empty, then, from equation Grad-OT with $\lambda = 0$, one can see that $y_j$ never appears in the update of $\theta$. As a consequence, nothing pushes $\mu_\theta$ to visit $y_j$.

The examples of Fig. 6 give an intuition of the behavior of WGAN learning algorithms when applied to high-dimensional datasets, such as images. First, depending on the choice of hyperparameters, the learning process may stabilize or not, as already noticed by (Stanczuk et al., 2021). Convergence issues do not prevent the learned distribution from realizing a good approximation of the data, in the sense that a non stabilized generator can produce plausible samples. The learned distribution is indeed able to fit a complex structure underlying the data, made of noisy curves that interpolate between datapoints. Samples obtained on the branch linking two datapoints $y_j, y_k$ can thus be seen as an interpolation between those two points. With images, such an interpolation of $y_j, y_k$ could be a blurry average ($L^2$ interpolation) or a better interpolation related to a geometric deformation of $y_j$ into $y_k$. This is what we observe in the next section when fitting a generative model to the MNIST database. However, considering the instability of the learning process and the number of parameters involved, it is difficult to ensure *a priori* that a generative model with a given architecture is appropriate for sampling accurately of a given dataset. As illustrated in experiments, such a model should be able to produce plausible interpolations of data points while exploiting the diversity of the dataset to avoid mode collapse.

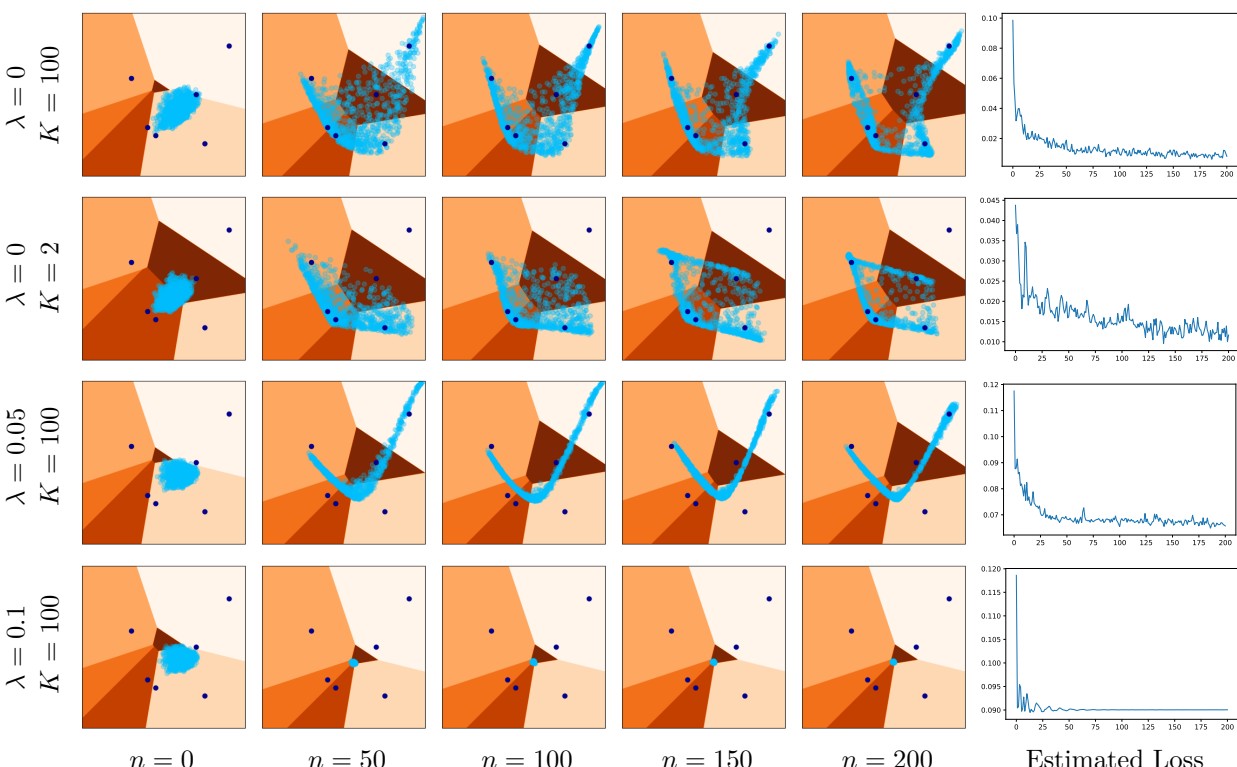

Figure 6: **Generator learning for** 6**-point dataset** $\mathcal{Y}$ **with MLP generator.** Results are displayed using the same presentation as in Figure 4. On the left we indicate the regularization parameter $\lambda$ and the number of iterations $K$ of the ASGD inner loop. The learning rate for $\psi$ is here fixed to $\gamma = 0.1$. See the text for comments.

### 5.3 Generator learning for MNIST dataset

We finally address the problem of digits generation by leaning a generative model on the MNIST database containing $J = 6 \cdot 10^4$ images. We discuss the behavior of the alternating Algorithm 1 and examine the impact of the regularization parameter $\lambda$ both on the visual results and the convergence of the loss function. We also discuss the effect of parameterizing the dual variable $\psi$ by a neural network.

The detailed settings used for these experiments are given in Appendix F.2. The generator parameters $\theta$ are optimized with ADAM algorithm (Kingma & Ba, 2015). The dual variable $\psi$ is optimized with Pytorch implementation of ASGD, with warm start initialization.

On Fig. 7, we display sampled digits obtained with the generative networks learned with Algorithm 1 run with different settings. The generators learned with unregularized OT ($\lambda = 0$) produce mostly convincing samples, slightly more blurry than the images of the database. Some of the samples generated do not look exactly like a digit but like a mixture of different digits. This reflects the fact that the generative network interpolates between the images of the database. The two tested architectures for the generator are MLP and DCGAN. They produce comparable results, with a slight advantage for DCGAN, which provides cleaner samples thanks to its more complex architecture adapted to two-dimensional data.

In all cases, the quality of visual results deteriorate when the regularization parameter $\lambda$ grows. For very small $\lambda$, the results are comparable to the unregularized case. For larger $\lambda$, the outputs of the generative networks concentrate on a blurry average of the database, due to the estimation bias of $W_\lambda$ (Genevay et al., 2018). This can be understood from the gradient formula (57) which involves the gradient of the regularized $c$-transform given by (47). When $\lambda \to +\infty$, $\nabla \psi^{c,\lambda}(x)$ degenerates to a simple average $\int_{\mathcal{Y}} \nabla_x c(x, y) d\nu(y)$. With the blur created by entropic regularization on the transport plan, the sampled points are pushed

towards all the target points in a mixed manner. In contrast, with unregularized transport, each sampled point $x$ is pushed towards the data point $T_\psi(x)$ assigned by the current OT map, as illustrated in Fig. 2.

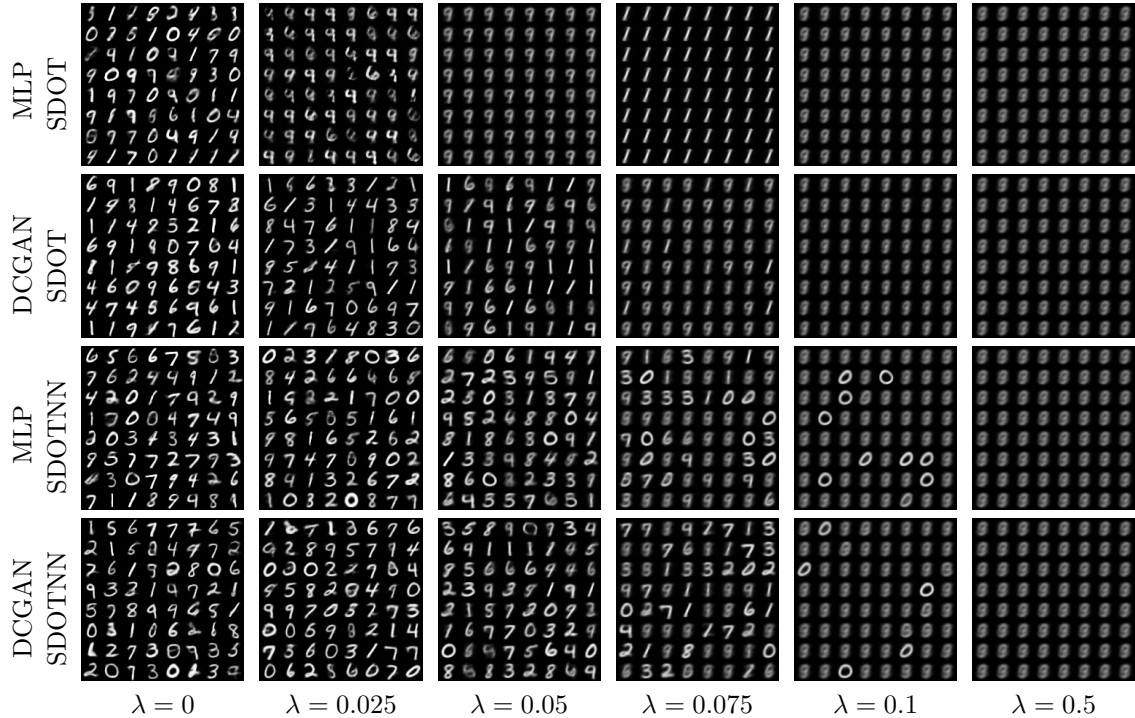

Figure 7: **Generator learning for MNIST dataset.** We compare here several generative networks trained with different architectures for the generator $g_\theta$ (MLP or DCGAN) and the dual variable $\psi$ with no parameterization (SDOT) in the two first rows, or MLP parameterization (SDOTNN) in the two last rows, and varying the parameter $\lambda$ of entropic regularization.

To better understand these visual results, we now examine the behavior of the loss function depending on the adopted setting. Fig. 8 shows the evolution of the estimated loss $H_\lambda(\underline{\psi}_\theta, \theta)$ along the iterates of Algorithm 1, when the dual variable $\psi$ is either parameterized as a vector in $\mathbb{R}^J$ (SDOT, for semi-dual OT) or a neural network (SDOTNN, for semi-dual OT with neural network).

The loss stabilizes, with small oscillations, in $\approx 500$ iterations, and the limit values obtained with both parameterizations (SDOT and SDOTNN) are similar. Note that the limit value is lower with the SDOTNN parameterization. Since the adopted multilayer perceptron has here $> 5 \cdot 10^5$ parameters (and is thus much larger than $J = 6 \cdot 10^4$), it is likely that any value $(\psi(y_j))_{1 \leqslant j \leqslant J} \in \mathbb{R}^J$ can be attained with such a parameterization for $\psi$. Notice also that the loss decreases in a more stable way with the SDOTNN parameterization. This parameterization is indeed likely to be more robust to the individual changes on $\psi(y_j)$ when updating the parameters $\theta$ of the generator, even when using warm start as done here.

The convergence speed does not improve drastically when using a larger regularization parameter $\lambda$, in contrast to what we observed with the Dirac generator of Section 5.2.1. This observation is confirmed in Fig. 9 where we display results obtained with various regularization parameters $\lambda$ and the four tested combinations of architectures for the generative network and the dual variable. These experiments illustrate that increasing the regularization parameter leads to a smoother optimized functional, which reflects in a more stable evolution of the loss. For very small $\lambda$ ($\leqslant 0.025$), we do not observe improvement of convergence speed with respect to $\theta$. In this low regularization regime, we suggest that the behavior of the loss evolution depends on the chosen architecture and the hyperparameters (step size strategies for ASGD or ADAM).

We finally display in Fig. 10, for fixed generator parameters $\theta$, the evolution of the loss $\psi \mapsto H_\lambda(\psi, \theta)$ (defined in (68)) during the inner ASGD loop used for optimizing the dual variable $\psi$. In order to complete

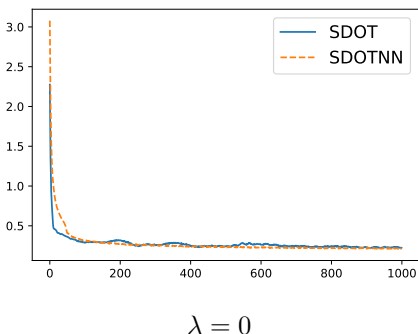 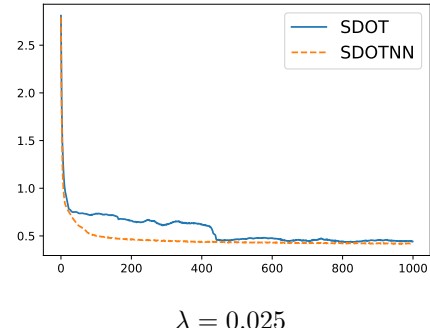

$$\lambda = 0 \qquad\qquad\qquad\qquad \lambda = 0.025$$

Figure 8: Evolution of the estimated loss $H_\lambda(\underline{\psi}_\theta, \theta)$ along the iterates of Algorithm 1 to learn a DCGAN generating MNIST digits. For each update of $\theta$, the loss is computed by using the current estimate $\underline{\psi}_\theta$ of the dual variable. For two values of the regularization parameter ($\lambda = 0$ on the left and 0.025 on the right), we compare the OT loss values obtained by parameterizing $\psi$ directly as a vector in $\mathbb{R}^J$ (SDOT) or as a neural network (SDOTNN). See the text for comments.

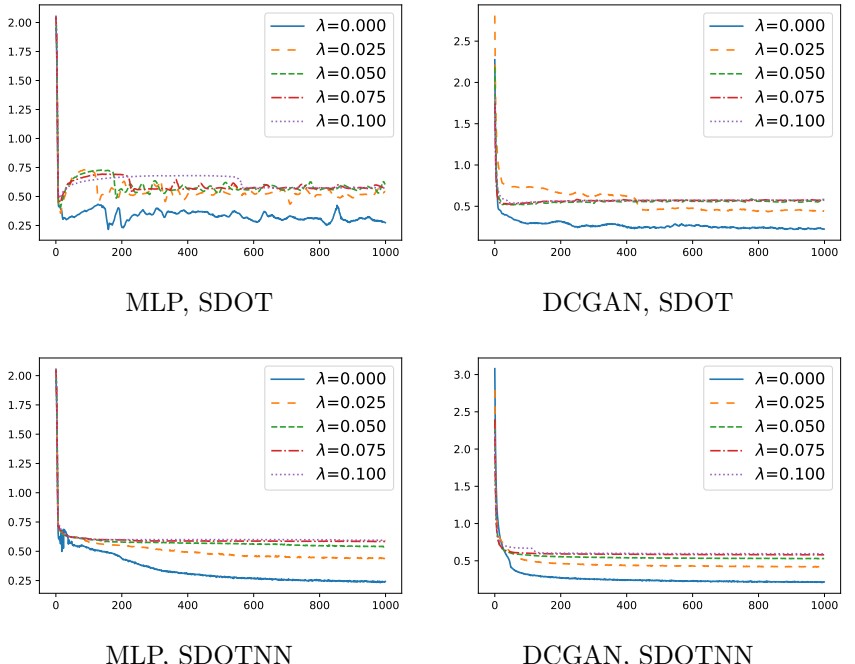

MLP, SDOT

DCGAN, SDOT

MLP, SDOTNN

DCGAN, SDOTNN

Figure 9: Evolution of the estimated loss $H_\lambda(\underline{\psi}_\theta, \theta)$ along the iterates of Algorithm 1, for the four tested combinations of parameterizations of the generator (MLP or DCGAN) and the dual variable (SDOT or SDOTNN). For each update of $\theta$, the loss is computed by using the current estimate $\underline{\psi}_\theta$ of the dual variable. Let us recall that the loss function $H_\lambda$ depends on the regularization parameter $\lambda$, which explains why the limit value attained by the algorithm actually increases when $\lambda \to 0$. See the text for additional comments.

the comparison, we also include the convergence plot obtained with the ADAM algorithm applied on the same problem. These convergence curves reflect again the slow convergence of the ASGD algorithm. We observed that a careful tuning of the learning rate of ASGD is necessary to obtain a sufficient decrease of the loss. Next, the convergence plots obtained with ASGD are similar with both parameterizations SDOT and SDOTNN. For small value of the regularization, turning to the ADAM algorithm does not improve the convergence speed for the SDOT parameterization. However, we remark that using the ADAM algorithm with the SDOTNN parameterization seems beneficial for all tested regularization parameters. The loss value obtained after 100 iterations is lower with SDOTNN than with SDOT, and the convergence is faster.

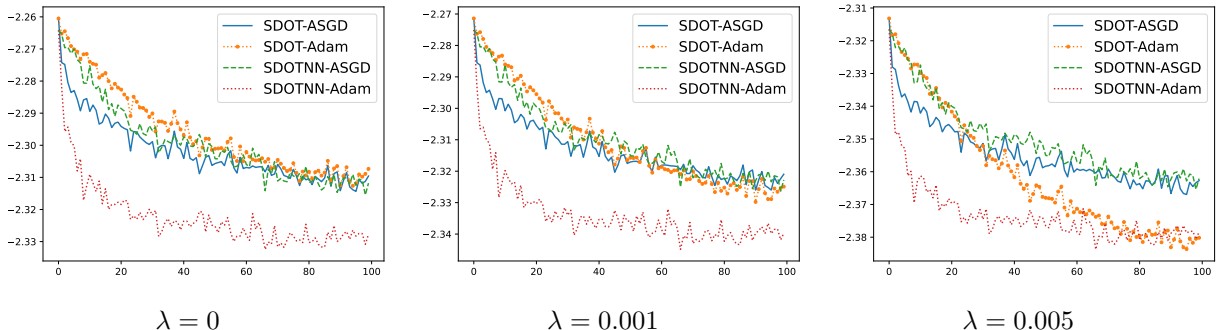

$\lambda = 0$         $\lambda = 0.001$         $\lambda = 0.005$

Figure 10: Evolution of the estimated loss $\psi \mapsto H_\lambda(\psi, \theta)$ along the iterates of the inner loop of Algorithm 1. Here, the parameters $\theta$ of the DCGAN generator is fixed, *i.e.* we consider a semi-discrete OT problem between a fixed $\mu_\theta$ and $\nu$. For several values of the regularization parameter, we compare the evolution of the loss when parameterizing $\psi$ directly as a vector in $\mathbb{R}^J$ (SDOT) or as a neural network (SDOTNN). For both parameterizations, the optimization is done using either ASGD with decreasing step size ($\frac{5}{\sqrt{k}}$ for SDOT and $\frac{0.1}{k^{0.8}}$ for SDOTNN) or ADAM (with learning rate 0.001). See the text for comments.

## 6 Conclusion

In this paper we give new insights on the theory and practice for learning generative networks with regularized Wasserstein distances. On the theoretical side, we prove a gradient formula for the minimized loss in two different frameworks: in the semi-discrete case (*i.e.* when the target distribution $\nu$ has finite support) without regularization, and in a more general case (with a general $\nu$) with entropic regularization. These results are based on weak regularity hypotheses on the cost and the generator, which are satisfied for many $\mathscr{C}^1$ cost functions and neural network generators with $\mathscr{C}^1$ activation functions. In the semi-discrete setting, it is also required that the generator does not charge the Laguerre interface, which is the generic case encountered in practice. These hypotheses are helpful to better understand the possible degenerate cases that can be encountered, and we provide such a counterexample.

On the practical side, we show that an alternating algorithm can approximate the solution of this optimization problem. The inner loop of this algorithm consists in approximating an optimal dual potential for regularized OT with a stochastic optimization algorithm. With experiments on a low-dimensional dataset, we illustrate that this alternating optimization algorithm has no reason to fall on the points of inexisting gradients, but that it may exhibit various singular behaviors. In most cases, it stabilizes to a generator which forms a relevant approximation of the target measure. Nevertheless, because of the fixed number of iterations for the inner loop, it can also oscillate or get trapped in sub-optimal positions (e.g. mode collapse on a single data point), depending on the choice of hyperparameters. Experiments on MNIST digits demonstrate that our algorithm is able to learn a neural network generating plausible images in a high-dimensional setting. Convincing visual results are indeed obtained with zero or small regularization parameter $\lambda$. We observe that the smoothing of the targeted loss function is not sufficient to significantly improve the convergence speed of the optimization algorithm estimating the generator parameters, while inducing a bias.

From this study, we claim that the success or failure of the WGAN learning process is not only affected by the chosen loss function (1-Wasserstein, 2-Wasserstein, Jensen-Shannon divergence, etc) but its also crucially depends on the choice of generator architecture and the choice of the optimization strategy.

Concerning the stochastic optimization used to solve the semi-dual OT problem, we observed that it may be beneficial in terms of convergence speed to parameterize the dual variable with a neural network, provided that one uses a well-chosen and carefully tuned algorithm for the optimization.

The main limitation of the considered algorithm is that the inner loop is based on the computation of a regularized *c*-transform. It thus requires, at each iteration, to visit all data points, in order to find a kind of biased nearest neighbor. In order to scale up to larger database, a possibility is to approximate

the regularized *c*-transform with a batch strategy (Mallasto et al., 2019a). As perspectives, it would be interesting to control the errors made at each iteration by the ASGD early-stop or by a batch strategy, in order to get a globally stable optimization process; or to consider other Stochastic Gradient Descent Ascent strategies (Sebbouh et al., 2022).

## Acknowledgments

This study has been carried out with financial support from the French Research Agency through the GOTMI, Mistic and PostProdLEAP projects (ANR-16-CE33-0010-01, ANR-19-CE40-005 and ANR-19-CE23-0027-01) and from the GdR ISIS through the Remoga project.

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

## A    A Neural Network Architecture validating Hypothesis ($\mathsf{G}_\Theta$)

In this section, we consider a neural network architecture given by a composition of affine layers and $\mathscr{C}^1$ Lipschitz activation functions. More precisely, the generator is defined as the composition of $L$ elementary layers defined by

$$\forall x \in \mathbb{R}^{d_{\ell-1}}, \quad f_{\ell,a_\ell,b_\ell}(x) = \rho_\ell(a_\ell x + b_\ell) , \tag{74}$$

where $a_\ell \in \mathbb{R}^{d_\ell \times d_{\ell-1}}, b_\ell \in \mathbb{R}^{d_\ell}$ and where $\rho_\ell : \mathbb{R}^{d_\ell} \to \mathbb{R}^{d_\ell}$ is a $\mathscr{C}^1$ and Lipschitz function. With a random input noise $Z$ in $\mathcal{Z} = \mathbb{R}^s$, and intermediate dimensions $s = d_0, d_1, \dots, d_L = d$, the generator is defined by

$$\forall \theta \in \Theta, \forall z \in \mathbb{R}^s, \quad g(\theta,z) = f_{L,a_L,b_L} \circ \cdots \circ f_{1,a_1,b_1}(z) , \tag{75}$$

where the parameters $\theta = (a_\ell, b_\ell)_{1 \leq \ell \leq L}$ are taken in an open set $\Theta \subset \prod_{\ell=1}^{L} \mathbb{R}^{d_\ell \times d_{\ell-1}} \times \mathbb{R}^{d_\ell}$.

The following proposition shows that Hypothesis $(\mathsf{G}_\Theta)$ is satisfied for such a neural network architecture.

**Proposition 3.** *Let us consider a generator $g$ of the previous form (75). Assume that the activation functions $\rho_1, \ldots, \rho_L$ are $\mathscr{C}^1$ and Lipschitz with the same constant $K_\rho$. Assume also that the input noise is integrable, that is, $\mathbb{E}[\|Z\|] < \infty$. Then $g$ satisfies Hypothesis $(\mathsf{G}_\Theta)$.*

*Proof.* Since the affine layers and activation layers are $\mathscr{C}^1$, it is clear that $\theta \mapsto g(\theta, Z)$ is $\mathscr{C}^1$. In order to show Hypothesis $(\mathsf{G}_\Theta)$, we only have to show that (22) holds on any compact subset of $\Theta$. Let $V \subset \Theta$ be a compact. Since $V$ is bounded, there is $\kappa > 0$ such that for all $\theta = (a_\ell, b_\ell) \in V$, we have $\|a_\ell\| \leqslant \kappa$ and $\|b_\ell\| \leqslant \kappa$. In particular, the corresponding function $f_{\ell, a_\ell, b_\ell}$ is $\kappa K_\rho$-Lipschitz.

Let $\theta = (a_\ell, b_\ell)_{1 \leq \ell \leq L} \in V$ and $\theta' = (a'_\ell, b'_\ell)_{1 \leq \ell \leq L} \in V$. First, assume that $\theta, \theta'$ only differ on one component: for a fixed $\ell$, let us decompose $\theta = (r_\ell, \theta_\ell, s_\ell), \theta' = (r_\ell, \theta'_\ell, s_\ell)$ with $r_\ell = (\theta_l)_{l < \ell}$ and $s_\ell = (\theta_l)_{l > \ell}$. We can then write

$$g(\theta, z) = R_{\ell, r_\ell}(f_{\ell, a_\ell, b_\ell}(T_{\ell, t_\ell}(z)) = R_{\ell, r_\ell}(\rho_\ell(a_\ell T_{\ell, t_\ell}(z) + b_\ell)) , \tag{76}$$

where $R_{\ell, r_\ell} = f_{L, a_L, b_L} \circ \cdots \circ f_{\ell+1, a_{\ell+1}, b_{\ell+1}}$ and $T_{\ell, t_\ell} = f_{\ell-1, a_{\ell-1}, b_{\ell-1}} \circ \cdots \circ f_{1, a_1, b_1}$. It is clear that $R_{\ell, r_\ell}$ and $T_{\ell, t_\ell}$ are respectively $(\kappa K_\rho)^{L-\ell}$ and $(\kappa K_\rho)^{\ell-1}$-Lipschitz. Therefore,

$$\|g(\theta', z) - g(\theta, z)\| \leq (\kappa K_\rho)^{L-\ell} \|\rho_\ell(a_\ell T_{\ell, t_\ell}(z) + b_\ell) - \rho_\ell(a'_\ell T_{\ell, t_\ell}(z) + b'_\ell)\| \tag{77}$$

$$\leq (\kappa K_\rho)^{L-\ell} K_\rho \|(a_\ell - a'_\ell) T_{\ell, t_\ell}(z) + b_\ell - b'_\ell\| \tag{78}$$

$$\leq (\kappa K_\rho)^{L-\ell} K_\rho (\|a_\ell - a'_\ell\| \|T_{\ell, t_\ell}(z)\| + \|b_\ell - b'_\ell\|). \tag{79}$$

It remains to bound $\|T_{\ell, t_\ell}(z)\|$. For that, let us remark that for any $l$,

$$\|f_{l, a_l, b_l}(x)\| \leq \|f_{l, a_l, b_l}(0)\| + \kappa K_\rho \|x\| = \|\rho_l(b_l)\| + \kappa K_\rho \|x\| \leq c_l + \kappa K_\rho \|x\|, \tag{80}$$

where $c_l = \|\rho_l(0)\| + \kappa K_\rho$. Thus, with a simple recursion, we get

$$\|g(\theta, z)\| \leq \sum_{l=1}^{L} (\kappa K_\rho)^{L-l} c_l + (\kappa K_\rho)^L \|z\|. \tag{81}$$

Since $T_{\ell, t_\ell}$ is the concatenation of only the $\ell - 1$ first layers, we have also

$$\|T_{\ell, t_\ell}(z)\| \leq C_\ell + (\kappa K_\rho)^{\ell-1} \|z\| \quad \text{with} \quad C_\ell = \sum_{l=1}^{\ell-1} (\kappa K_\rho)^{\ell-1-l} c_l. \tag{82}$$

Gathering the previous inequalities, we get

$$\|g(\theta', z) - g(\theta, z)\| \leq \left( C_\ell (\kappa K_\rho)^{L-\ell} K_\rho + (\kappa K_\rho)^{L-1} K_\rho \|z\| \right) \|a_\ell - a'_\ell\| + (\kappa K_\rho)^{L-\ell} K_\rho \|b_\ell - b'_\ell\|. \tag{83}$$

Upper bounding $\|a_\ell - a'_\ell\|$ and $\|b_\ell - b'_\ell\|$ by $\|\theta - \theta'\|$, we finally get, if $\theta, \theta'$ differ only on the $\ell$-component,

$$\forall z \in \mathbb{R}^s, \quad \|g(\theta', z) - g(\theta, z)\| \leq (D_\ell + D'_\ell \|z\|) \|\theta - \theta'\| \quad \text{with} \quad \begin{cases} D_\ell = (1 + C_\ell)(\kappa K_\rho)^{L-\ell} K_\rho \\ D'_\ell = (\kappa K_\rho)^{L-1} K_\rho \end{cases}. \tag{84}$$

Therefore,

$$\forall \theta, \theta' \in V, \ \forall z \in \mathbb{R}^s, \quad \|g(\theta', z) - g(\theta, z)\| \leq K(z) \|\theta - \theta'\| \quad \text{with} \quad K(z) = \sum_{\ell=1}^{L} D_\ell + D'_\ell \|z\|, \tag{85}$$

and we have $\mathbb{E}[K(Z)] < \infty$ because $\mathbb{E}[\|Z\|] < \infty$. $\qquad\square$

## B   Counter-example with regularized OT

We consider the same measures than in Proposition 2: $\mu_\theta = \delta_\theta$ and $\nu = \frac{1}{2}\delta_{y_1} + \frac{1}{2}\delta_{y_2}$. For $\lambda > 0$, one has

$$F_\lambda(\psi, \theta) = \psi^{c,\lambda}(\theta) = -\lambda \log\left(\frac{1}{2}e^{-\frac{c(\theta,y_1)-\psi(y_1)}{\lambda}} + \frac{1}{2}e^{-\frac{c(\theta,y_2)-\psi(y_2)}{\lambda}}\right), \tag{86}$$

and from the primal formulation, one can also get directly that

$$h_\lambda(\theta) = \frac{1}{2}c(\theta, y_1) + \frac{1}{2}c(\theta, y_2). \tag{87}$$

For any $\psi$, the function $\theta \mapsto F_\lambda(\psi, \theta)$ is differentiable on $\mathbb{R}^d$ for $p > 1$ and on $\mathbb{R}^d \setminus \{y_1, y_2\}$ for $p = 1$. Thus in the regularized case $\lambda > 0$, $F_\lambda(\psi, \cdot)$ has no singularity on the interface $A_\psi$. For a fixed $\theta_0$, the maximization of $H(\cdot, \theta_0)$ leads to the same solutions satisfying (27) than in Proposition 2, *i.e.* any $\psi_{*0}$ such that

$$e^{-\frac{c(\theta,y_1)-\psi(y_1)}{\lambda}} = e^{-\frac{c(\theta,y_2)-\psi(y_2)}{\lambda}}. \tag{88}$$

It follows that Formula (Grad-OT) holds in that case:

$$\nabla h_\lambda(\theta_0) = \frac{1}{2}\nabla_x c(\theta, y_1) + \frac{1}{2}\nabla_x c(\theta, y_2) = \nabla_\theta F_\lambda(\psi_{*0}, \theta_0). \tag{89}$$

Let us also notice that using regularized OT does not allow to cope with all differentiability issues. Indeed, for $p = 1$, $h_\lambda$ is not differentiable at $y_1, y_2$ even for $\lambda > 0$. This illustrates the need of a regularity hypothesis on the cost, even in the case of regularized OT.

## C   Continuity of $c$-transforms

In this section, we will recall properties of regularized $c$-transforms. For that, we need a modulus of continuity of the cost function, that is, the smallest function $\omega$ such that

$$\forall x, x' \in \mathcal{X}, \ \forall y, y' \in \mathcal{Y}, \quad |c(x,y) - c(x',y')| \leqslant \omega\left(\|x - x'\| + \|y - y'\|\right). \tag{90}$$

Since $c$ is continuous on the compact $\mathcal{X} \times \mathcal{Y}$, it is uniformly continuous, thus $\lim_{\delta \to 0} \omega(\delta) = 0$.

**Lemma 6** ((Santambrogio, 2015; Feydy et al., 2019)). *For $\lambda \geqslant 0$, any $c$-transform $\psi^{c,\lambda}$ has a modulus of continuity which is bounded by the modulus of continuity of the cost function.*

*Proof.* If $u \leqslant v$ holds pointwise, then $\operatorname{softmin} u \leqslant \operatorname{softmin} v$ pointwise. Also, for a constant $k \in \mathbb{R}$, $\operatorname{softmin}(k + u) = k + \operatorname{softmin}(u)$. But, from the definition of $\omega$, we have $c(x,y) - \psi(y) \leqslant \omega(\|x - x'\|) + c(x',y) - \psi(y)$. By taking the soft-min, we thus obtain $\psi^{c,\lambda}(x) \leqslant \omega(\|x - x'\|) + \psi^{c,\lambda}(x')$, and by symmetry, this leads to

$$|\psi^{c,\lambda}(x) - \psi^{c,\lambda}(x')| \leqslant \omega(\|x - x'\|). \tag{91}$$

$\square$

**Lemma 7.** *For $\lambda \geqslant 0$, and any $\psi, \chi \in \mathscr{C}(\mathcal{Y})$, $\|\psi^{c,\lambda} - \chi^{c,\lambda}\|_\infty \leqslant \|\psi - \chi\|_\infty$. In other words, the map $\psi \mapsto \psi^{c,\lambda}$ is 1-Lipschitz for the uniform norm.*

*Proof.* Applying the monotonicity of the softmin operation to the inequality $c(x,y) - \psi(y) \leqslant \|\psi - \chi\|_\infty + c(x,y) - \chi(y)$, we get $\psi^{c,\lambda}(x) \leqslant \|\psi - \chi\|_\infty + \chi^{c,\lambda}(x)$. On can conclude with a symmetry argument. $\square$

The following lemma states that the optimal dual potentials vary continuously with respect to the input measure as soon as they are unique up to additive constants.

**Lemma 8** ((Feydy et al., 2019)). *Assume that $\mathcal{X}, \mathcal{Y}$ are compact and $c$ is continuous. Let us fix $x_0 \in \mathcal{X}$. Assume that $\nu$ is fixed, and that $\mu_n$ converges weak-$\star$ to $\mu$ in the space of probability measures on $\mathcal{X}$. Assume that the Kantorovich potentials associated with $W_\lambda(\mu, \nu)$ are unique up to an additive constant, which is the case if $\lambda > 0$. For each $n$, let $\varphi_n$ be a c-concave Kantorovich potential for $W_\lambda(\mu_n, \nu)$ such that $\varphi_n(x_0) = 0$ and let $\varphi$ be the (necessarily c-concave) Kantorovich potential for $W_\lambda(\mu, \nu)$ such that $\varphi(x_0) = 0$. Then $\varphi_n \to \varphi$ uniformly on $\mathcal{X}$.*

*Proof.* By compactness of $\mathcal{X} \times \mathcal{Y}$, $c$ has a bounded modulus of continuity $\omega(\delta)$ which tends to zero when $\delta \to 0$. By (91), we obtain that the functions $|\varphi_n|$ are bounded by $\sup_{x \in \mathcal{X}} \omega(\|x - x_0\|) < \infty$. Besides, Lemma 6 also shows that the functions $\varphi_n$ are uniformly equicontinuous on $\mathcal{X}$. Therefore, Arzela-Ascoli theorem ensures that the family $\{\varphi_n, n \in \mathbb{N}\}$ is relatively compact in $\mathscr{C}(\mathcal{X})$.

Now, assume, by contradiction, that $(\varphi_n)$ does not tend to $\varphi$ in $\mathscr{C}(\mathcal{X})$. Then there would exist $\varepsilon > 0$ and a subsequence $(\varphi_{r(n)})$ such that

$$\|\varphi_{r(n)} - \varphi\|_\infty > \varepsilon \quad \forall n. \tag{92}$$

By relative compactness, one can then extract a subsequence $(\varphi_{r(s(n))})$ which converges in $\mathscr{C}(\mathcal{X})$ to a function $\tilde{\varphi}$. Using the monotonicity of soft-min, this implies that $(\varphi_{r(s(n))}^{c,\lambda})$ also converges in $\mathscr{C}(\mathcal{X})$ to $\tilde{\varphi}^{c,\lambda}$. Thus

$$W(\mu_n, \nu) = \int \varphi_n d\mu + \int \varphi_n^{c,\lambda} d\nu \xrightarrow[n \to \infty]{} \int \tilde{\varphi} d\mu + \int \tilde{\varphi}^{c,\lambda} d\nu. \tag{93}$$

Finally, we also have $W(\mu_n, \nu) \to W(\mu, \nu)$ since $\mu_n \overset{*}{\to} \mu$. Thus $W(\mu, \nu) = \int \tilde{\varphi} d\mu + \int \tilde{\varphi}^{c,\lambda} d\nu$ and $\tilde{\varphi}$ is a Kantorovich potential for $W(\mu, \nu)$ and $\tilde{\varphi}(x_0) = \lim \varphi_n(x_0) = 0$. Using the uniqueness assumption, we get $\tilde{\varphi} = \varphi$ which contradicts (92). $\square$

## D   Technical Results

We first recall the proof of the envelope theorem (Oyama & Takenawa, 2018, Prop. A.1).

**Theorem 7** (Envelope theorem Oyama & Takenawa (2018)). *Let $X$ be a topological space. Let $A$ be an open set of a normed vector space $E$. Let $f : X \times A \to \mathbb{R}$ be a function and let us denote*

$$\forall a \in A, \quad v(a) = \sup_{x \in X} f(x, a). \tag{94}$$

*Let $s : A \to X$ be such that for all $a \in A$, $v(a) = f(s(a), a)$. Let $\alpha \in A$ be a point such that*

- *$s$ is continuous at $\alpha$,*

- *the partial differential $D_a f$ of $f$ with respect to $a$ exists in a neighborhood of $(s(\alpha), \alpha)$, and is continuous at $(s(\alpha), \alpha)$.*

*Then $v$ is differentiable at $\alpha$ and $Dv(\alpha) = D_a f(s(\alpha), \alpha)$.*

*Proof.* Let $\xi = s(\alpha)$ and let $\varepsilon > 0$. The second hypothesis gives an open neighborhood $U \times V$ of $(\xi, \alpha)$ in $X \times A$ such that for any $(x, a) \in U \times V$, $f(x, \cdot)$ is differentiable at $a$ and such that the partial differential $(x, a) \mapsto D_a f(x, a)$ is continuous on $U \times V$. By continuity of $s$, $s^{-1}(U) \cap V$ is an open neighborhood of $\alpha$ and thus it exists $\eta > 0$ such that $\|h\| < \eta$ implies $\alpha + h \in V$ and $s(\alpha + h) \in U$.

By definition of $v$, we have for any $h \in E$ such that $\|h\| < \eta$,

$$f(\xi, \alpha + h) - f(\xi, \alpha) \leqslant v(\alpha + h) - v(\alpha) \leqslant f(s(\alpha + h), \alpha + h) - f(s(\alpha + h), \alpha). \tag{95}$$

On the one hand, by definition of $D_a f(\xi, \alpha)$, there exists $\eta_1 \in (0, \eta)$ such that $\|h\| < \eta_1$ implies

$$|f(\xi, \alpha + h) - f(\xi, \alpha) - D_a f(\xi, \alpha)h| \leqslant \varepsilon \|h\|. \tag{96}$$

On the other hand, for $\|h\| < \eta$, $t \in [0, 1] \mapsto f(s(\alpha + h), \alpha + th)$ is differentiable on $[0, 1]$ and therefore, by virtue of the mean value theorem, there exists $\theta_h \in (0, 1)$ such that

$$f(s(\alpha + h), \alpha + h) - f(s(\alpha + h), \alpha) = D_a f(s(\alpha + h), \alpha + \theta_h h) h. \tag{97}$$

By continuity of $D_a f$, there is an open neighborhood $\bar{U} \times \bar{V} \subset U \times V$ of $(\xi, \alpha)$ such that

$$\forall (x, a) \in \bar{U} \times \bar{V}, \quad \|D_a f(x, a) - D_a f(\xi, \alpha)\| \leqslant \varepsilon, \tag{98}$$

Again, by continuity of $s$, $s^{-1}(\bar{U}) \cap \bar{V}$ is an open neighborhood of $\alpha$ and thus, there exists $\eta_2 \in (0, \eta)$ such that $\|h\| < \eta_2$ implies $\alpha + h \in s^{-1}(\bar{U}) \cap \bar{V}$. Therefore, for $\|h\| < \eta_2$, $(s(\alpha + h), \alpha + \theta_h h) \in \bar{U} \times \bar{V}$ and thus

$$|f(s(\alpha+h), \alpha+h) - f(s(\alpha+h), \alpha) - D_a f(\xi, \alpha) h| \leqslant \|D_a f(s(\alpha+h), \alpha + \theta_h h) - D_a f(\xi, \alpha)\| \|h\| \leqslant \varepsilon \|h\|. \tag{99}$$

Finally, for $\|h\| < \min(\eta_1, \eta_2)$ we get $|v(\alpha + h) - v(\alpha) - D_a f(\xi, \alpha) h| \leqslant \varepsilon \|h\|$, which proves that $v$ is differentiable at $\alpha$ and $D_a v(\alpha) = D_a f(\xi, \alpha)$. $\qquad \square$

The next proposition gives the support of a push-forward distribution.

**Proposition 4.** *Let $Q = [-1, 1]^s$, and $g : Q \to \mathbb{R}^d$ continuous. Let $Z$ be a random variable with uniform distribution $\zeta$ on $Q$ and let $\mu = g \sharp \zeta$ be the distribution of $g(Z)$. Then the support of $\mu$ is exactly $g(Q)$.*

*Proof.* Since $g$ is continuous, $g(Q)$ is compact and closed. Thus $U = g(Q)^c$ is open, and we have

$$\mu(U) = \mathbb{P}(g(Z) \in U) = \zeta(g^{-1}(U)) = 0, \tag{100}$$

because $g^{-1}(U)$ does not intersect $Q$. This proves that $\text{Supp}(\mu) \subset g(Q)$. Now, if $V$ is an open set such that $\mu(V) = 0$, then $\mathbb{P}(Z \in g^{-1}(V)) = 0$, which gives $g^{-1}(V) \cap Q = \varnothing$ because $g^{-1}(V)$ is open. It follows that $V \subset g(Q)^c$, which proves that $\text{Supp}(\mu)$ is exactly $g(Q)$. $\qquad \square$

## E  Proof of Theorem 6

Using Theorem 5, we only have to study the regularity of $\theta \mapsto W_\lambda(\mu_\theta, \mu_\theta)$ for $\lambda > 0$. We will use again the envelope theorem recalled in Appendix D. We consider the set $C$ of $\mathscr{C}^1$ and $L$-Lipschitz functions on $\mathcal{X}$ equipped with the norm $\|\chi\|_\infty + \|\nabla \chi\|_\infty$ and we use the dual expression (66) which can be written with the integral operator $I$ of (17) and with

$$\mathcal{F}_\lambda(\chi, \eta, \theta) = I(\chi, \theta) + I(\eta, \theta) + \lambda - \lambda E_\lambda(\chi, \eta, \theta) \tag{101}$$

$$\text{where} \quad E_\lambda(\chi, \eta, \theta) = \mathbb{E}\left[\exp\left(\frac{\Gamma(\chi, \eta, \theta, Z, W)}{\lambda}\right)\right] \tag{102}$$

$$\text{and} \quad \Gamma(\chi, \eta, \theta, z, w) = \chi(g_\theta(z)) + \eta(g_\theta(w)) - c(g_\theta(z), g_\theta(w)), \tag{103}$$

where $W, Z$ are two independent random variables of distribution $\zeta$. For any $\chi \in C$, differentiating under the integral as in Lemma 5 gives that $I(\chi, \cdot)$ is differentiable on $\Theta$ with gradient

$$\nabla_\theta I(\chi, \theta) = \mathbb{E}\left[D_\theta g(\theta, Z)^T \nabla \chi(g(\theta, Z))\right]. \tag{104}$$

By the same reasoning, one obtains that $E_\lambda(\chi, \eta, \cdot)$ is differentiable on $\Theta$ with gradient

$$\nabla_\theta E_\lambda(\chi, \eta, \theta) = \mathbb{E}\left[\frac{\nabla_\theta \Gamma(\chi, \eta, \theta, Z, W)}{\lambda} \exp\left(\frac{\Gamma(\chi, \eta, \theta, Z, W)}{\lambda}\right)\right] \tag{105}$$

$$\text{with} \quad \nabla_\theta \Gamma(\chi, \eta, \theta, z, w) = D_\theta g(\theta, z)^T \left(\nabla \chi(g(\theta, z)) - \nabla_x c(g(\theta, z), g(\theta, w))\right) \tag{106}$$

$$+ D_\theta g(\theta, w)^T \left(\nabla \eta(g(\theta, w)) - \nabla_y c(g(\theta, z), g(\theta, w))\right). \tag{107}$$

If $P_0$ is any bounded set of $C$, using a bound similar to (56) and the dominated convergence theorem shows that $(\chi, \theta) \mapsto I(\chi, \theta)$ is continuous on $P_0 \times V$. Similarly, one can see that $\Gamma$ is Lispchitz in $\theta$ with a $(Z, W)$-integrable bound

$$|\Gamma(\chi, \eta, \theta, z, w) - \Gamma(\chi, \eta, \tau, z, w)| \leqslant (\|\nabla \chi\|_\infty K(z) + \|\nabla \eta\|_\infty K(w) + L) \|\theta - \tau\|. \tag{108}$$

Since $\Gamma$ is also bounded by $\|\chi\|_\infty + \|\psi\|_\infty + \|c\|_\infty$ and since the continuity of $\Gamma$ with respect to $(\chi, \eta, \theta)$ can be deduced from (103), this is enough to show that $(\chi, \eta, \theta) \mapsto \nabla_\theta E_\lambda(\chi, \eta, \theta)$ is continuous on $P_0 \times V$.

As we have a continuous selection of optimal dual variables $(\chi, \eta)$ and as the gradient $\nabla_\theta \mathcal{F}_\lambda(\chi, \eta, \theta)$ exists and is continuous in $(\chi, \eta, \theta)$, all the conditions required by the envelope theorem are satisfied. Thus we can differentiate under the max in (66)

$$\nabla_\theta W_\lambda(\mu_\theta, \mu_\theta) = \nabla_\theta \mathcal{F}_\lambda(\chi_*, \eta_*, \theta), \tag{109}$$

where $(\chi_*, \eta_*)$ is a pair of Kantorovich potentials for $W_\lambda(\mu_\theta, \mu_\theta)$. Finally, by symmetry, there exists $k \in \mathbb{R}$ such that $\chi_* = \eta_* + k$ and by definition $E_\lambda(\chi_*, \eta_*, \theta) = 1$, so that $\mathcal{F}_\lambda(\chi_*, \eta_*, \theta) = 2I(\chi_*, \theta) - k$ and therefore

$$\nabla_\theta \mathcal{F}_\lambda(\chi_*, \eta_*, \theta) = 2\nabla I(\chi, \theta) = 2\mathbb{E}\left[D_\theta g(\theta, Z)^T \nabla \chi_*(g(\theta, Z))\right]. \tag{110}$$

Putting all terms together, we get the desired formula for $\nabla_\theta s_\lambda(\theta)$.

# F    Detailed Settings used for Experiments on generator learning

## F.1    Detailed Settings for the experiments with synthetic datasets

We here gather the parameters and network architectures used in the experiments of Section 5.2.

- $N = 50$ iterations on $\theta$ with SGD algorithm (with constant learning rate $\eta = 0.2$), except for the MLP generator which is optimized with Adam (with constant learning rate 0.05)

- $K = 100$ iterations of the inner loop with ASGD algorithm (see learning rates below) with cold start initialization (unless otherwise specified)

- The cost $c(x, y)$ is the quadratic cost $\|x - y\|^2$.

- For the generator we consider various architectures $g_\theta$ and input noise $Z$:
  - a Dirac generator $g_\theta^{\mathrm{dirac}}(z) = \theta$ parameterized by $\theta \in \mathbb{R}^2$,
  - a disk generator with fixed radius $r > 0$ with $Z$ a uniform random variable on $[0, 1]^2$, parameterized by $\theta \in \mathbb{R}^2$:
    $$g_\theta^{\mathrm{disk}}(z) = \theta + r\sqrt{z_1}(\cos(2\pi z_2), \sin(2\pi z_2)).$$
  - an ellipse generator (referred to as 1-ellipse) with $Z$ a uniform random variable on $[0, 1]^2$, parameterized by $\theta = (t, a, b) \in \mathbb{R}^2 \times \mathbb{R} \times \mathbb{R}$:
    $$g_\theta^{\mathrm{ell}}(z) = t + \sqrt{z_1}(a\cos(2\pi z_2), b\sin(2\pi z_2)).$$
  - a mixture of two ellipses (denoted by 2-ellipse generator) with $Z$ a uniform random variable on $[0, 1]^3$, parameterized by $\theta = (\theta_1, \theta_2) \in (\mathbb{R}^2 \times \mathbb{R} \times \mathbb{R})^2$:
    $$g_\theta^{\mathrm{mixell}}(z) = \begin{cases} g_{\theta_1}^{\mathrm{ell}}(z_1, z_2) & \text{if} \quad z_3 < \frac{1}{2} \\ g_{\theta_2}^{\mathrm{ell}}(z_1, z_2) & \text{otherwise.} \end{cases}$$
  - a multilayer perceptron (MLP) with $Z$ a uniform random variable on $[0, 1]^{\mathsf{din}}$, $\mathsf{nhid}$ fully-connected layers with dimension $\mathsf{dhid}$ and activation function $\mathsf{Elu}(1)$. In Section 5.2.2, the layers are set to $\mathsf{din} = 5$, $\mathsf{dhid} = 10$ and $\mathsf{nhid} = 3$. In Appendix G, the parameters are indicated in the captions of the figures.

- All batches of $Z$ are made of $b = 100$ samples, except for the Dirac generator where 1 sample suffices.

- The dual variable $\psi$ is modeled by a vector $\psi \in \mathbb{R}^J$.

- The dual variable $\psi$ is optimized with the ASGD algorithm written in (71). The step size strategy is $\frac{\gamma}{k^{0.5}}$ with learning rate $\gamma > 0$ (values indicated in the text).

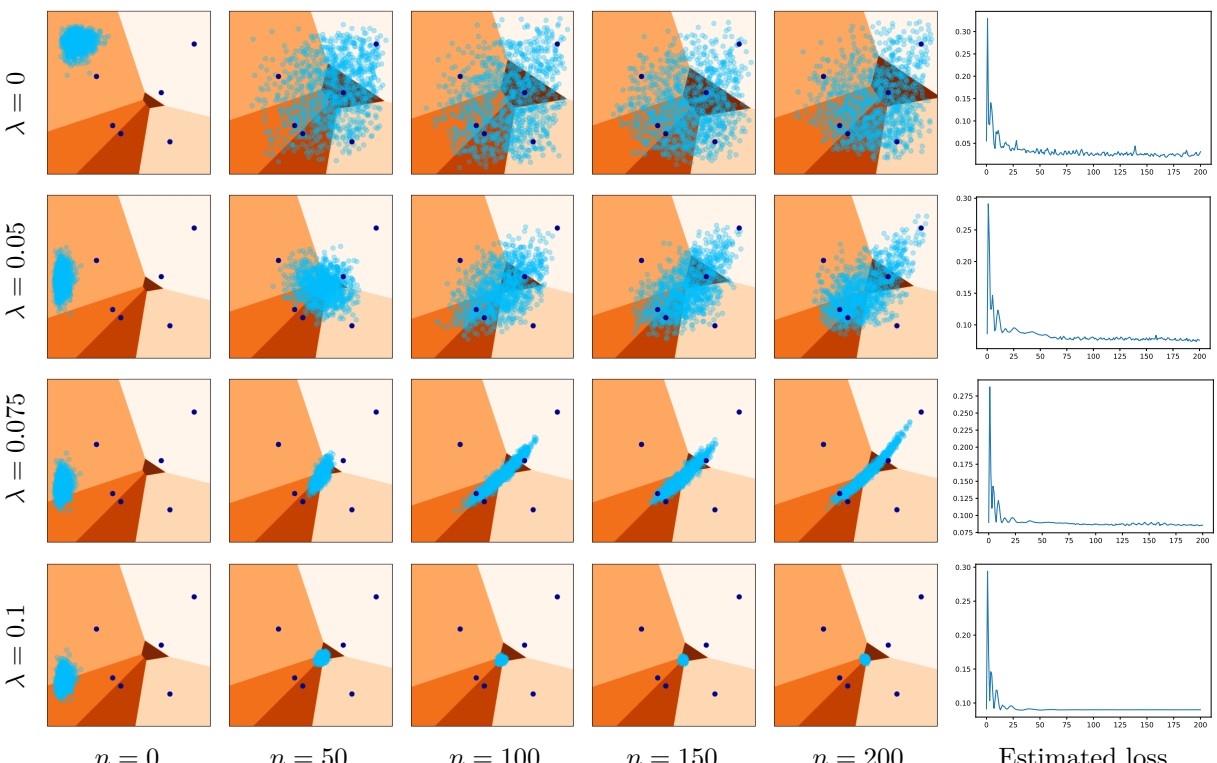

Figure 11: **Generator learning for** $6$**-point dataset** $\mathcal{Y}$ **with MLP generator.** The parameters of the MLP are din $= 100$, dhid $= 256$, nhid $= 3$. In the first columns, along the number of iterations $n$, we display $\mathcal{Y}$ (dark blue) and the current position of $\mu_\theta$ (sampled points in light blue). The Laguerre diagram associated to the current value of $\underline{\psi}_\theta$ is displayed in the background with the colored partition. The last column shows the estimated loss $H_\lambda(\underline{\psi}_\theta, \theta)$ along the iterates. On the left we give the regularization parameter $\lambda$.

## F.2 Detailed Settings for the MNIST experiment

We here gather the parameters and network architectures used in the experiments of Section 5.3.

- $N = 3000$ iterations on $\theta$ with ADAM algorithm Kingma & Ba (2015) with learning rate $0.001$

- $K = 10$ iterations of the inner loop with ASGD algorithm with warm start initialization

- The cost $c(x, y)$ is the quadratic cost $\alpha^{-1}\|x - y\|^2$ normalized by $\alpha = \frac{1}{J}\sum_{y \in \mathcal{Y}} \|y\|^2$

- For the generator $g_\theta$ we consider two different architectures:

  - a multilayer perceptron (MLP) with four fully-connected layers; the number of channels for the successive hidden layers is $256, 512, 1024$ with activation functions LeakyReLU($0.2$).
  - a Deep Convolutional GAN (DCGAN) (Radford et al., 2016) adapted for the dimension $28 \times 28$ of MNIST images, with 4 deconvolution layers; the number of channels for the hidden layers are $256, 128, 64$.

- The input of these generators is a random variable $Z$ following the uniform distribution on $[-1, 1]^{100}$. All batches of $Z$ are made of $b = 200$ samples.

- The dual variable $\psi$ is either directly modeled by a vector $\psi \in \mathbb{R}^J$, or parameterized by a multilayer perceptron with four fully-connected layers; the number of channels for the successive hidden layers is $512, 256, 128$. These two different settings are respectively referred to as SDOT (for semi-dual OT) and SDOTNN (for semi-dual OT with neural network).

- The dual variable $\psi$ is optimized with the Pytorch implementation of ASGD (`torch.asgd`), which has parameters `lr` and `alpha`. The step size strategy for the ASGD inner loop has been chosen as:
  - for the parameterization SDOT, $\texttt{lr} = 5, \texttt{alpha} = 0.5$,
  - for the parameterization SDOTNN, $\texttt{lr} = 0.1, \texttt{alpha} = 0.8$.

  The other parameters are fixed to default values.

## G Additional experiments on $6$-point dataset

In this appendix we provide additional results of generative model learning on the 6-point dataset.

In Fig. 11, for a fixed large MLP architecture, we vary the regularization parameter $\lambda$.

In Fig. 12, we fix the regularization parameter $\lambda = 0$ and we vary the configuration of the MLP architecture.

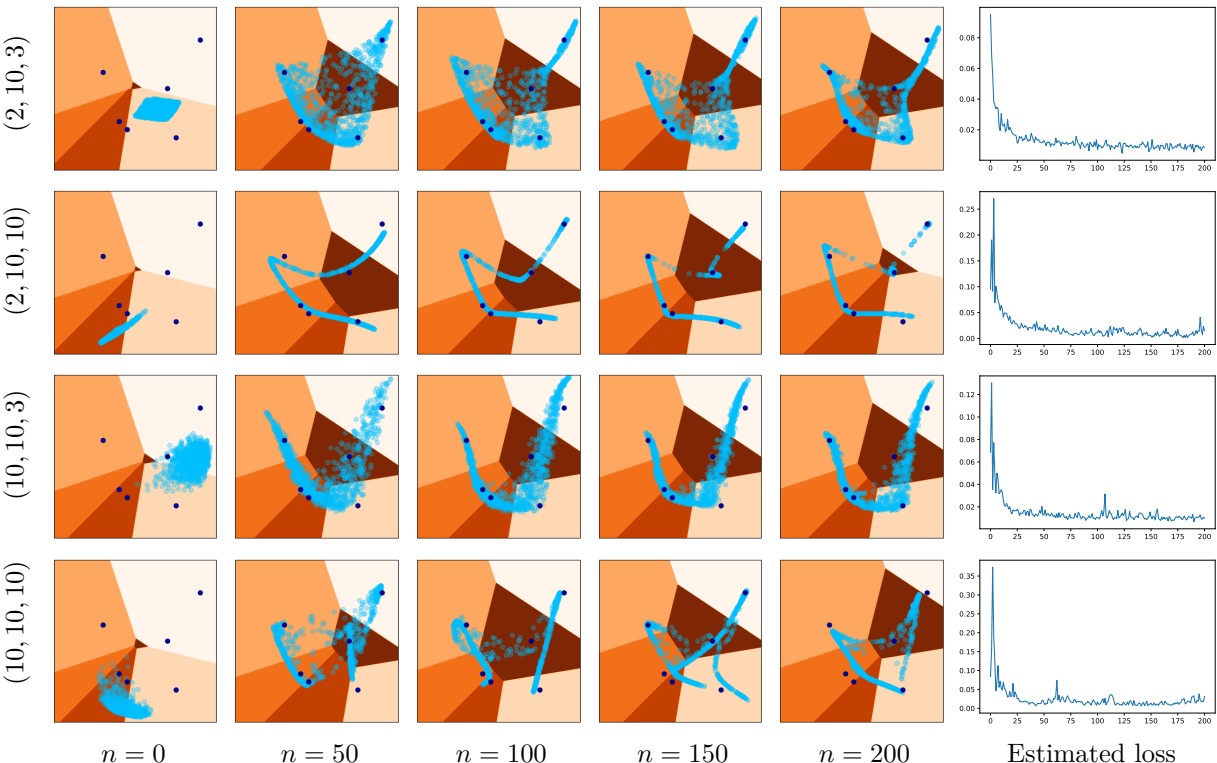

Figure 12: **Generator learning for $6$-point dataset $\mathcal{Y}$ with MLP generator.** On the left we indicate the tuple of parameters $(\mathsf{din}, \mathsf{dhid}, \mathsf{nhid})$. In the first columns, along the number of iterations $n$, we display $\mathcal{Y}$ (dark blue) and the current position of $\mu_\theta$ (sampled points in light blue). The Laguerre diagram associated to the current value of $\underline{\psi}_\theta$ is displayed in the background with the partition indicated in colors. The last column shows the estimated loss $H_\lambda(\underline{\psi}_\theta, \theta)$ along the iterates. The regularization parameter is set to $\lambda = 0$.

