# OpenReview forum: "On the Gradient Formula for learning Generative Models with Regularized Optimal Transport Costs"
_TMLR — Accepted by TMLR_

### Review · Reviewer_GVxx · 2022-08-27

**Summary Of Contributions:**

The paper approaches the problem of generative modeling. Namely, the authors consider the optimal transport formulation of the problem. That is, given the target measure $\nu$, one tries to minimize the Wasserstein distance
$$W(\mu_\theta,\nu) = \inf_{\pi} \int c(x,y) d\pi (x,y),$$
where $\pi$ is the joint distribution with marginals $\mu_\theta$ and $\nu$, w.r.t. parameters $\theta$ of the parametric model $\mu_\theta$. In the general case, the objective includes entropy regularization, which is the KL-divergence between $d\pi(x,y)$ and $d\mu_\theta(x)\cdot d\nu(y)$. Further, the regularized problem is reformulated to the semi-dual formulation (proposed in [1])
$$W(\mu_\theta,\nu) = \sup_{\psi} \int \psi^{c,\lambda}(x) d\mu_\theta(x) + \int\psi(y)d\nu(y),$$
where $\psi$ is a continuous function and $\psi^{c,\lambda}$ is its regularized c-transform. Note that the parameter $\lambda$ is the entropy regularization coefficient.

The paper's main contribution is the development of rigorous conditions on the differentiability of $W(\mu_\theta,\nu)$ (in the semi-dual formulation) w.r.t. $\theta$. Besides the theoretical results, the authors study the role of $\lambda$ and the convergence of the algorithm empirically.

The theoretical part of the paper is structured as follows. First, the authors give the motivational example for the case when the considered objective is not differentiable (section 2.5). In more details, the authors parameterize the model $\mu_\theta$ as a delta-function as point $\theta$, i.e., $\delta(x-\theta)$. Motivated by this example, the authors prove Theorems 3 and 5 for $\lambda = 0$ (unregularized transport) and $\lambda > 0$ (regularized transport) correspondingly. The main tool for the proof is the envelope theorem. This theorem defines the differentiability conditions for the function
$$f(\alpha) = \sup_x g(x,\alpha)$$
w.r.t. $\alpha$. After defining the sufficient conditions, the formula for the gradient is obtained simply by the chain rule.

The empirical study presented in the paper is not an immediate consequence of the theoretical developments. For the generative model $\mu_\theta$, the authors consider a neural network, which is then optimized via ADAM using the gradient of the semi-dual formulation. To find the dual function $\psi$, the authors perform optimization in the inner loop and consider two cases: $\psi \in \mathbb{R}^J$, and $\psi$ is parameterized by a neural network. Note that in the case $\psi \in \mathbb{R}^J$, this algorithm boils down to the algorithm proposed in [2].

[1] Genevay, Aude, Marco Cuturi, Gabriel Peyré, and Francis Bach. "Stochastic optimization for large-scale optimal transport." Advances in neural information processing systems 29 (2016).

[2] Liu, Dong, Minh Thành Vu, Saikat Chatterjee, and Lars K. Rasmussen. "Entropy-regularized optimal transport generative models." In ICASSP 2019-2019 IEEE International Conference on Acoustics, Speech and Signal Processing (ICASSP), pp. 3532-3536. IEEE, 2019.

**Broader Impact Concerns:**

The paper doesn't discuss the broader impact and doesn't have to, in my opinion.

**Requested Changes:**

To address my main concern, I would suggest discussing the practical implications of the established sufficient conditions supported by corresponding empirical study. To be concrete,
- based on the motivational example (section 2.5) describe which parameterizations of $\mu_\theta$ (potentially practical) lead to non-differentiable functions;
- discuss the x-regularity condition of the cost function giving examples for practitioners;
- discuss the family of algorithms satisfying the hypothesis $G_\theta$.

If some conditions are too strict and since the proposed set of conditions is not a criterion, I would suggest discussing the gap between sufficient and necessary conditions.

I would also suggest fixing the minor issues from the list. However, they are not critical for acceptance.

**Strengths And Weaknesses:**

The paper approaches an important problem of generative modeling via optimal transport. The semi-dual formulation proposed in [1] has a broad range of applications, and the rigorous outline of its differentiability (the contribution of the current paper) has both theoretical and practical implications. I've gone through all the derivations, and the proofs are correct to the best of my knowledge. However, I'm not an expert in the optimal transport field, and I might have skipped some subtle mistakes. For the empirical part of the paper, the authors base their claims only on the empirical facts presented in the paper.

In my opinion, the main weakness of the paper is the absence of immediate practical implications. First, the motivational example is not very practical; I'm not aware of any problem where the considered parametrization would be helpful. Second, the authors establish a set of sufficient conditions for the differentiability of the objective, but they don't discuss how one can satisfy these conditions in practice. For Theorem 3, the authors discuss how to relax the x-regularity condition and discuss that the potential uniqueness (Remark 2) condition can be discarded in the case of regularized transport. However, the x-regularity condition appears again in Theorem 5 (regularized transport), and the condition on $G_\Theta$ is not discussed at all. After all, it is not clear whether these conditions can be satisfied in practice for any relevant model and data. As a consequence, the empirical part of the paper seems to be unattached to the theoretical part and reads like an algorithm from [2] but with a neural network.

The list of minor issues:
- I find the proof of the motivational example (Proposition 2) not completely clear. Namely, I don't understand why the whole space $\mathcal{Y}$ is restricted to two points $y_1,y_2$ while performing the c-transform in (33). Do we always restrict $\mathcal{X}$ and $\mathcal{Y}$ to the support of distributions? I haven't found this detail in the paper.
- Throughout the paper, the authors use the nabla operator, not defining the differentiation argument, which might be confusing (although it's still possible to infer from the context). I think it will add a bit of clarity if the authors simply mention in the notation section that this notation means differentiation w.r.t. the immediate input of the function.
- Typo on page 9. "the c-transform (12)" -> "the c-transform (11)"
- page 10. Instead of "By definition of Laguerre cells," I think the authors really mean "from (40)".
- equation (43) is confusing. As follows from the proof of Lemma 5, the differentiation in (43) is performed only w.r.t. the first argument of $c(x,T_\psi(x))$. However, in the formula, the differentiation is w.r.t. $x$, which also means differentiation w.r.t. the second argument in this case.
- equation (54) $\nabla_x c(g_\theta(z), y)$. I guess this is the differentiation w.r.t. the first argument.
- the plots would benefit from axis labels.
- there is a typo in conclusion: "not charge".

---

> ### Author Response · Authors · 2022-11-09
> **Reponse to Reviewer GVxx (Part 2)**
>
> **About the domain of c-transforms**
>
> It is clear that the primal formulation of optimal transport on $\mathcal{X} \times \mathcal{Y}$ can always be restricted to the true supports $\mathcal{X'}, \mathcal{Y'}$ of the distributions $\mu, \nu$ respectively. We agree that it is less obvious on the dual formulation, and we will add a remark to clarify this point.
>
> To be more precise, the original problem only involves integrals with respect to transport plans $\pi \in \Pi(\mu, \nu)$, and is thus not affected to what happens on $\mu$-negligible (resp. $\nu$-negligible sets). Nevertheless, the definition of the $c$-transform
> $$ \forall x \in \mathcal{X}, \quad \psi^c(x) = \inf_{y \in \mathcal{Y}} c(x,y) - \psi(y) $$
> involves an infimum on **all** points of $\mathcal{Y}$, as emphasized in Remark 5.5 of [Villani, "Optimal Transport, Old and New",2008].
>
> However, the dual formulation of optimal transport does not change if the dual variables $\varphi, \psi$ are restricted to the support of $\mu, \nu$ respectively. In order to give some intuition on that, let us consider the semi-discrete case where $\mathcal{Y}$ is a finite set with $J$ points (as in Section 3 of our paper). Then, the semi-dual formulation of optimal transport writes
> $$ W(\mu, \nu) = \max_{\psi \in \mathbb{R}^J} \int_{\mathcal{X}} \psi^c d\mu + \sum_{y \in \mathcal{Y}} \nu(\\{y\\}) \psi(y)  .$$
> Suppose that $y_0 \in \mathcal{Y}$ is such that $\nu(\\{y_0\\})=0$. Then, starting from an arbitrary $\psi$, we can modify $\psi(y_0)$ by decreasing it towards $-\infty$: this modification only increases $\psi^c$ (and thus $\int \psi^c d\mu$) while not changing the second term $\sum \nu(\\{y\\}) \psi(y)$. And taking $\psi(y_0) \to -\infty$ is then equivalent to exlude $y_0$ in the infimum that defines $\psi^c$. This shows that the restriction of the infimum to the true support of $\nu$ does not impact the semi-dual formulation.
>
> **About the connection with Liu et al. "Entropy-regularized optimal transport (...)", 2019**
>
> This is true that the algorithm studied in our experimental part lies close to other already-published algorithms, and we will discuss that more carefully.
>
> In order to be as close as possible to the WGAN objective criterion, our experimental study differs in various aspects in comparison to the one of [Liu et al., 2019], namely:
> - We do not resort to batch sampling on the dataset (the c-transforms are computed on all data points).
> Such a mini-batch optimization strategy, as studied thoroughly in [Fatras et al., "Learning with minibatch Wasserstein : asymptotic and gradient properties", AISTATS, 2020], is equivalent to an implicit regularization which does not enable to compute the exact transportation cost and yields less sharp results.
> - In our optimization scheme, the optimization of dual variables can be accelerated using warm start from previous iterations, while the mini-batch optimization of [Liu et al., 2019] requires to solve a different optimal transport problem at each iteration.
> - We aim at better understanding the role of the regularization parameter $\lambda$, and in particular compare regularized and unregularized transport.
> - In our experiments, the squared Euclidean distance is used as the ground cost function $c$.
> In [Liu et al., 2019], a feature transform is trained jointly with the generative model, likewise to GANs in which the discrimitive neural network in GANs is a classifier whose first layers can be interpreted as a feature transform.
>
>
>
> **Minor issues**
>
> In order to address the suggestion of Reviewer ndPu about notations, we will add an appendix devoted to notations and conventions.
> Also, in the "Notations" paragraph of Section 2, we added a remark about the convention used for diffentials and gradients. For univariate function, we simply use $\nabla$ without confusion. For multivariate functions, we always add an index to indicate which argument is used for differentiation.
>
> In particular, for the case of Equation (43), $\nabla_x c$ indeed refers to the gradient of $c$ with respect to its first argument.

---

> ### Author Response · Authors · 2022-11-09
> **Reponse to Reviewer GVxx (Part 1)**
>
> **About the motivational example of Proposition 2**
>
> We agree that the case described in Proposition 2 is a toy example.
> Actually, our Theorem 3 shows that this failure case does not happen for most of the practical generators of class $C^1$ (see the responses below about the regularity hypotheses). Indeed, its most constraining hypothesis is that $\mu_\theta$ does not charge the union $A_\psi$ of boundaries of Laguerre cells. For example, if $c(x,y) = \\|x-y\\|$ or $\\|x-y\\|^2$ on $\mathbb{R}^d$, then this hypothesis will be true as soon as $\mu_\theta$ has a density with respect to the Lebesgue measure.
>
> However, we can modify our example of Proposition 2 in order to be a little bit more practical while escaping this hypothesis. Recall that in this example, with $Y = \\{y_1,y_2\\}$ and a discrete generated distribution  $\mu_\theta = \delta_\theta$, the optimal transport cost $h$ is differentiable for any $\theta \not \in Y$ while the dual expression $H(\psi_\star,.)$ is not differentiable at the interface $\mathcal{I}$ of the Laguerre cells, corresponding to the bisector of $[y_1, y_2]$ defined by $\psi_\star$. For instance, considering the semi-discrete problem where $\mu_\theta$ is a one-dimensional distribution supported on a segment which is orthogonal to $[y_1, y_2]$ yields the same differentiability problem.
> We will add a diagram after Proposition 2 to illustrate the problem of gradient discontinuity at the interface of Laguerre cells.
>
>
> One can then think of a more general version of this example, where one tries to fit a low-dimensional model $\mu_\theta$ (supported on a sub-manifold of Lebesgue measure 0) to an empirical data distribution $\nu$. One of the main insight of our paper is that the diffentiability problem is avoided during optimization as soon as the generative model is positioned not to charge the boundaries of Laguerre cells. However, in the exceptional case where it charges the boundaries of Laguerre cells, a differentiability problem is to be expected with positive probability.
>
> This example is actually close to the practical case of the Fig. 1 of [Chen et al., "A gradual, semi-discrete approach to generative network training via explicit wasserstein minimization.", ICML, 2019] where the authors try to fit a one-dimensional generator to data lying on a circle.
>
> **About the x-regularity condition**
>
> We agree to give more discussion on the x-regularity condition.
>
> To sum up for practical applications:
> - the quadratic cost $c(x,y) = \\|x-y\\|^2$ satisfies the x-regularity condition,
> - the cost $c(x,y) = \\|x-y\\|$ does not satisfy it as soon as $\mathcal{X}, \mathcal{Y}$ are not disjoint.
>
> Since $c(x,y) = \\|x-y\\|$ appears for any application related to the $W^1$ cost (and thus for most Wasserstein GANs), we proposed Theorem 4 in order to include this case, at the cost of a small additional assumption on $\mu_\theta$.
>
> Also, the x-regularity condition appears again in our results of Section 4 on regularized optimal transport. Actually, the entropic regularization is not sufficient to avoid all regularity problems. In order to illustrate this point, we will add a remark that describes how the example of Proposition 2 adapts to the case of regularized optimal transport. In this case, the value of the functional
> $$h_\lambda(\theta) = \frac{1}{2} c(\theta, y_1) + \frac{1}{2} c(\theta, y_2) $$
> still holds for $\lambda > 0$. Therefore, the singular points of $\theta \mapsto c(\theta, y_j)$ may still produce singular points of $h_\lambda$.
>
>
> **About the regularity hypothesis for the generator $G_\theta$**
>
> We will add a remark to comment on this hypothesis. This remark shows that the hypothesis is satisfied when $g$ is $C^1$ with input noise $Z$ supported on a compact. In particular, it is satisfied with a neural network with $C^1$ activation functions and input noise supported on a compact.

---

### Review · Reviewer_H5qL · 2022-09-11

**Summary Of Contributions:**

The paper proposes a complete set of hypotheses that ensure the validity of
the gradient formula which commonly appears in WGAN learning. Specifically, the authors consider the semi-discrete case (which appears in most WGANs - generator generates some distribution, data distribution is empirical) and additionally consider the general entropy-regularized case. They study the gradient formulas in these two cases. Also, for the (biased, not Sinkhorn) entropy-regularized case, the authors propose an algorithm to learn WGAN and test it on the grayscale MNIST dataset.

**Broader Impact Concerns:**

None.

**Requested Changes:**

I think the authors should improve their paper in terms of separation of their contribution and existing results (see the previous section comments).

From the experimental point of view, I think it might be better to study WGAN convergence with the unbiased entropy cost = Sinkhorn (see the reasons above), especially taking intro account the fact that the author provide a theoretical discussion of it, its potentials and gradient.

Also, it would be nice to see some experiment which is relevant to the “inexisting gradient issue”, e.g., demonstrating that the WGAN does not converge in this case. In other words, is the inexisting gradient an issue in practice, e.g., in some toy example?

Additionally, I suggest adding a discussion of the above-mentioned related works.


**Strengths And Weaknesses:**

The main strength of the paper is the detailed theoretical study of existence of the gradient of the optimal transport objectives which are used to train WGANs. The paper studies both the regularized case (semi-discrete) and entropic case and discusses conditions under which these gradients exist. At the same time, I think a noticable weak side of the paper is its writing. In the text, there are a lot of existing theoretical results (from Santambrogio's book, Genevay's papers etc.) and they are mixed together with the new results of this paper. It is not clear what is really new, how significant is the contribution w.r.t. the existing part. While there are some sections trying to describe this (e.g., Section 2.4), I have to admit that I got lost and can not fully access the amount of the theoretical contribution of this paper. Overall, I miss a short section (somewhere at the beginning) summarizing in a sufficiently detailed form the main theorems/statements which the authors claim as their contribution.

I do not understand the point of experimental evaluation in this paper. In particular, using Entropic Wasserstein loss in GANs (lambda>0) leads to biased solution, i.e., W_{\lambda}(\mu_{\theta}, v) is minimal not necessarily when \mu_{\theta}=\nu. Thus, the minimizer in entropic WGAN is not necessarily the data distribution \vu. Probably this is what we see in Figure 1 (the bias of the optimal solution increases?). I think this issue can be alleviated by considering Sinkhorn divergences (a.k.a. unbiased entropic transport) which the authors discuss in Section 4.4. However, their algorithmic methodology and is entirely about biased Wasserstein. I do not think this evaluation (or their proposed WGAN learning methodology) adds any value to the paper.

Since the paper is mostly about existence/inexistence of objective gradient, a relevant experimental exaluation would be to show what are the potential pitfalls of the inexistent gradient in practice. So far I see the opposite: the case when this gradient may not exist (lambda=0) leads to the best generative model.

The paper misses discussion of several related works [1,2,3,4] where the authors also study some relevant aspects of WGAN training. In particular, the authors make several unsupported claims which I am not sure are correct. For example, "When learning a Wasserstein generative model, the performance of the dual solver used for estimating the OT cost (3) is therefore a key point." -- the above-cited works mostly dismiss this claim.

In the unregularized case (lambda=0), it is widely known that the OT gradient of the potential is related to the OT map and with mild assumptions they can be recovered one from another. In particular, there are a lot of works studying the semi-discrete optimal transport formulations and OT maps -- their methodology is very similar to what the authors discuss with the Laguerre cells [5]. I wonder how the presented results are related to semi-discrete OT in general. Thus, I think including some deeper discussion about semi-discrete OT methods is desired.

Minor
It would be nice to see some explanatory image in the section about semi-discrete OT.
section 1.3: inclucing -> including


[1] Mallasto, A., Montúfar, G., & Gerolin, A. (2019). How well do WGANs estimate the wasserstein metric?. arXiv preprint arXiv:1910.03875.

[2] Korotin, A., Li, L., Genevay, A., Solomon, J. M., Filippov, A., & Burnaev, E. (2021). Do neural optimal transport solvers work? a continuous wasserstein-2 benchmark. Advances in Neural Information Processing Systems, 34, 14593-14605.

[3] Stanczuk, J., Etmann, C., Kreusser, L. M., & Schönlieb, C. B. (2021). Wasserstein GANs work because they fail (to approximate the Wasserstein distance). arXiv preprint arXiv:2103.01678.

[4] Korotin, A., Kolesov, A., & Burnaev, E. (2022). Kantorovich Strikes Back! Wasserstein GANs are not Optimal Transport?. arXiv preprint arXiv:2206.07767.

[5] Lei, N., Su, K., Cui, L., Yau, S. T., & Gu, X. D. (2019). A geometric view of optimal transportation and generative model. Computer Aided Geometric Design, 68, 1-21.

---

> ### Author Response · Authors · 2022-11-09
> **Response to Reviewer H5ql (Part 2)**
>
> **About related works**
>
> We thank the reviewers for the suggested references [1-5] that we will include in our section "Related Works". Indeed, there have been many interesting studies on the relevance of using Wasserstein costs (with various dual solvers) in GAN learning.
>
> Our methodology uses the "c-transform" method mentioned in the suggested references [1] Mallasto et al. (2019) and [3] Stanczuk et al. (2021), except that we compute the c-transform on all data points of $\nu$ and not on a batch, which helps to stabilize learning.
> We will add the following paragraph in Section 1.3 to clarify that:
>
>
> > There have been several works (Mallasto et al., 2019; Stanczuk et al. 2021; Korotin et al., 2021, 2022) that question the performance of OT dual solvers and its impact on WGAN learning. (Mallasto et al., 2019) compared several algorithms to estimate the dual variables, namely WGAN-WC, WGAN-GP and two methods denoted  "$c$-transform" and ''$(c,\epsilon)$-transform'' (those two methods being similar to the algorithm we use here, but with a batchwise computation of the $c$-transform). (Mallasto et al., 2019) showed that these two last methods, while better estimating the OT cost, do not improve the visual quality of the generative model when used in WGAN learning (producing very blurry images for the CelebA database). These findings were confirmed by (Stanczuk et al., 2021) who gave more explanations related to biased estimation of Wasserstein gradients, false Wasserstein minima supported on barycenters, and the limitations of the Euclidean distance computed on natural images. Our experimental study shows that, with the simpler MNIST database, a WGAN learning based on $c$-transform computations on all data points (and not batchwise), can produce sharp images while relying on a provably-convergent estimation algorithm for the Wasserstein cost. Also, our theoretical analysis of gradients allows to better understand the impact of the errors made by the OT dual solver: the performance of the dual solver is not directly related to the sharpness of generated images, but more on the insurance that the generated distribution covers the whole database.
>
>
> If we understood correctly [2,4] by Korotin et al., the OT solvers compared therein are all based on various parameterizations of the dual problem with neural networks, whose optimization requires to draw some batches $\mu$ **and** $\nu$ (which are absolutely continuous). Thus we cannot directly compare to those solvers since we always use all points of $\nu$ (which is discrete in our experiments). Also for our experiments on MNIST digits generation, the groundtruth OT map (or cost) is not available, which hinders the computation of the same error measures than in [2,4].
> However, one can remark that the concept of $c$-transform is also central in the construction proposed in [4] where the MinFunnels are $c$-transforms attached to a finite set of points $(a_n)$ (and and one can easily imagine the Laguerre diagram associated with the MinFunnel function drawn in [4, Fig. 7]).
>
> **About related work on semi-discrete OT**
>
> We agree to include more discussion on the connection with previously-known results about semi-discrete optimal transport, and in particular the reference [5]. We will also add a diagram to illustrate the semi-discrete optimal transport map, and how it relates to the update of the generator $g_\theta$. This will also draw a connection with [4, Fig. 5] which displays the transported point that is used for the update of the generator.
>
> **About the performance of dual solvers**
>
> Indeed, we should be more careful in our claim on the performance of dual solvers. Your remark is interesting, and we think that the results of Section 3 of our paper sheds some light on this point: what happens if we use the value of $\psi$ which is not optimal? In other words, we can examine the update of $\theta$ when using the wrong $\psi$ in the $\nabla_\theta$-descent step on
> $$F(\psi, \theta) = \mathbb{E}[\psi^c(g_\theta(Z))].$$
> If $\psi$ is not optimal, the Laguerre diagram is not positioned correctly, and thus, with our Lemma 5, one can understand that descending on $\nabla \psi^c$ pushes $g_\theta(Z)$ towards a wrong point $T_\psi(g_\theta(Z))$. However, this point $T_\psi(g_\theta(Z))$ is still a data point, and thus descending likewise will still help $g_\theta$ to learn some characteristics of the data points. From this calculation, one can understand that the performance of the dual solver is important not for generating data-like points but for the generative network to cover the whole database (in a way that respects the mass of the empirical distribution $\nu$).
>
> These observations will be gathered in a remark at the end of Section 3.

---

> ### Author Response · Authors · 2022-11-09
> **Response to Reviewer H5ql (Part 1)**
>
>
> **About the organization of the paper and the highlight of our main results**
>
> We agree that the organization of the first version of the paper could be confusing with a mix between well-established results and our contributions. As you suggest, in Section 1.3 "Contributions and Outline", we now explain in more details our contributions and point to our main results (which are Proposition 2, Theorem 3, 4, 5, 6 and the experimental Section 6).
>
> Also, we will move Section 2.2 "Continuity of c-transforms" in Appendix. Recalling these elementary results is necessary to help the reader grasp the main concepts of our paper. Nevertheless the length of such information in the paper indeed adds confusion with respect to the presentation of our own results.
>
> We could either just move most of our proofs in Appendix, if the reviewers judge it preferable.
>
>
> **About the Sinkhorn divergence**
>
> It seems to us that you raise a delicate question about the Sinkhorn divergence, which, indeed, is known to avoid bias in practical learning problems with entropy-regularized optimal transport.
>
> In Section 4.4 we extended our theoretical result to the case of the Sinkhorn divergence. Since the question was asked several times by colleagues in the field, and since we were able to prove it, we prefered to include this theoretical result in the paper.
>
> However, it is true that this result is not used in practice in our paper. Actually, in our experimental study, we rather adopt a "semi-discrete" point of view, where $\mu_\theta$ has a density and $\nu$ is discrete. To the best of our knowledge, in this case, the evaluation of the Sinkhorn divergence $S_\lambda (\mu_\theta, \nu)$ is not straightforward because it requires the evaluation of $W_\lambda(\mu_\theta, \mu_\theta)$ which corresponds to a continuous optimal transport problem.
>
> We could have considered directly a batch-Wasserstein cost $W_{\lambda}(\hat{\mu_{\theta}}, \nu)$ relying on a batch version $\\hat{\\mu_{\\theta}}$ of $\\mu_{\\theta}$. But this batch procedure is also known to induce a bias (see [Fatras et al., "Learning with minibatch Wasserstein : asymptotic and gradient properties", AISTATS, 2020]). The debiasing process with the Sinkhorn divergence would be relative to the chosen batch.
>
> For these reasons, extending our experiments to the Sinkhorn divergence did not appear straightforward. But if we missed something about that, we are still open to the reviewers' suggestions to help us include this case.
>
>
> **About the pitfalls of inexisting gradients**
>
> Our main Theorem 3 shows that the failure case does not happen for most of the practical generators (even for lambda=0). Indeed, its most constraining hypothesis is that $\mu_\theta$ does not charge the union $A_\psi$ of boundaries of Laguerre cells. For example, if $c(x,y) = \\|x-y\\|$ or $\\|x-y\\|^2$ on $\mathbb{R}^d$, then this hypothesis will be true as soon as $\mu_\theta$ has a density with respect to the Lebesgue measure.
>
> For this reason, except in particular situations (see our response to Reviewer GVxx "About the motivational example of Proposition 2"), one cannot expect a practical pitfall from the gradient problem that we highlighted in this paper.
>
> However, in our experimental study, we aim at illustrating that, even in the unregularized case $\lambda = 0$, almost surely the algorithm never falls on a singular point, and thus the learning process continue harmlessly (and, as you emphasize, leading to the best visual results for unregularized transport). As you suggest, we will add a subsection in the experimental Section 6 in order to illustrate the behavior of the alternate optimization algorithm applied to the toy example of Proposition 2. This new subsection will also help to connect the theoretical results and the practical study.
>
> We will add a remark in the introductive Section 1.3 on our contributions, in order to better explain the positioning of our experimental study.

---

### Review · Reviewer_ndPu · 2022-10-30

**Summary Of Contributions:**

The paper considers entropy regularized and un-regularized OT formulation for learning generative models through GANs. The work proves formulas for obtaining gradient of the generative model in this formulation under some regularity assumptions.

**Broader Impact Concerns:**

No applicable

**Requested Changes:**

Could you please provide updates/answers corresponding to the points mentioned above.

**Strengths And Weaknesses:**

Strength:
- There is definitely a need to better understand the learning procedure and how/under what condition current algorithms work (or do not work). This paper is definitely a good step in that direction from a mathematical perspective.
Weaknesses:
- Writing:
  - The paper is notation heavy and I believe the presentation of the paper could be greatly improved with an introductory section on notations.
  - The related work section could be moved to the end of the paper or changed in a way that it is easier to follow with notations/contexts provided in the introduction.
- Regularity conditions:
  - I am still a bit uncertain about the differences between this work and previous works; Are there any concrete cases that this work shows the gradient exists/does not exist where the previous works cannot provide. If so, are there any practical implications?
  - Moreover, it is very difficult to gain intuitions on the regularity conditions. I am assuming that these regularity conditions are not verifiable in practice? If so, I believe this should be indicated in the paper.
  - Regarding Proposition 2, could the authors provide more intuition? For example, they could provide some insight on the intuitions behind this example. I am also wondering what happens when the objective is regularized? Would the gradient relationship still fail?
- Approximating the gradient:
  - As the authors have mentioned, it is impossible to exactly compute the inner dual solutions. One thing that is missing from the paper is the relationship between the approximations of the dual solution, and approximations of the gradient. This would be necessary for any practical algorithm to perform on this problem.
- Experiments:
  - The experimental section is a bit detached from the rest of the paper. For example, based on the experiments is there any practical reason why one might want to use regularization if it does not improve performance or speed of convergence?
  - As the authors have mentioned, increasing regularization introduces biases in approximating the generative distribution. Previous works have shown that one can overcome these by using methods such as Sinkhorn divergences. I am curious what the results would look like with those approaches.
  - Regarding the SDOTNN scenario, is it possible to change the number of parameters (compared to the number of data points) to see how/at which point does it affect the performance.

---

> ### Author Response · Authors · 2022-11-09
> **Response to Reviewer ndPu (Part 2)**
>
> **About Proposition 2 and its intuition**
>
> We will add a diagram to illustrate the problem of gradient discontinuity at the interface of Laguerre cells.
>
> Also, we will add a remark on the fact that for $p>1$, the gradient relationship (Grad-OT) holds true in the regularized case ($\lambda > 0$).
>
>
> **About approximating the gradient**
>
> The link between the $\theta$-gradient approximation and the dual solver for optimal transport is also raised by  Reviewer H5ql. We think that the results of Section 3 of our paper sheds some light on this interesting question.
>
> What happens if we use the value of $\psi$ which is not optimal when computing the $\theta$-gradient? We can examine the update of $\theta$ when using the wrong $\psi$ in the $\nabla_\theta$-descent step on
> $$F(\psi, \theta) = \mathbb{E}[\psi^c(g_\theta(Z))].$$
> If $\psi$ is not optimal, the Laguerre diagram is not positioned correctly, and thus, with our Lemma 5, one can understand that descending on $\nabla \psi^c$ pushes $g_\theta(Z)$ towards a wrong point $T_\psi(g_\theta(Z))$. However, this point $T_\psi(g_\theta(Z))$ is still a data point. Descending likewise will still help $g_\theta$ to learn some characteristics of the data points. From this calculation, one can understand that the performance of the dual solver is important not for generating data-like point but for the generative network to cover the whole database (in a way that respect the mass of the empirical distribution $\nu$).
>
> These observations will be gathered in a remark at the end of Section 3.
>
> **About the experimental section**
>
> - Our experiments indicate that there is no practical benefit in using regularized optimal transport ($\lambda > 0$) for WGAN learning with this alternate algorithm. However, we have not investigated batchwise strategy where regularized optimal transport could improve the convergence speed on discrete subproblems.
>
> - About the Sinkhorn divergence: in our experimental study, we rather adopt a "semi-discrete" point of view, where $\mu_\theta$ has a density and $\nu$ is discrete. To the best of our knowledge, in this case, the evaluation of the Sinkhorn divergence $S_\lambda (\mu_\theta, \nu)$ is not straightforward because it requires the evaluation of $W_\lambda(\mu_\theta, \mu_\theta)$ which corresponds to a continuous optimal transport problem. Here also, batchwise formulations could extend more easily to the Sinkhorn divergence, but batching also induces a bias, see [Fatras et al., Learning with minibatch Wasserstein : asymptotic and gradient properties, AISTATS, 2020].
>
> - We will work on the proposed experiment to see the impact of the number of parameters in the SDOTNN scenario.

---

> ### Author Response · Authors · 2022-11-09
> **Response to Reviewer ndPu (Part 1)**
>
> **About notations**
>
> In order not to break the flow of discussion of the paper, and in order to solve the reviewer's concern about notation, we propose to add an Appendix that summarizes the notation used in the paper.
>
> **About related works**
>
> We agree to give more details in the "Related works" section (see also the response to Reviewer H5qL on this point). We will also better single out our contributions in Section 1.3.
>
> The current location of the "Related works" section was chosen in order to give a fast overview of other papers that directly relate to our study. We fear that postponing this section and increasing it with more notations would add too many details which are not essential for the exposition of our paper. However, we could do it if all reviewers judge it necessary.
>
>
> **About regularity conditions and difference with previous works**
>
> The previous result on the gradient formula (Grad-OT) written by Arjovsky et al. (2017) assumes that both sides of the inequality exists, but does not give precise conditions to ensure existence of both sides. In order to answer the reviewer's suggestion (i.e. finding concrete cases where this formula fails), we need to find cases of non-existence of one side, which is exactly the point of our Proposition 2.
>
> Our main Theorem 3 shows that the failure of the gradient formula (Grad-OT) does not happen for most of the practical generators (even for lambda=0). Indeed, its most constraining hypothesis is that $\mu_\theta$ does not charge the union $A_\psi$ of boundaries of Laguerre cells. For example, if $c(x,y) = \\|x-y\\|$ or $\\|x-y\\|^2$ on $\mathbb{R}^d$, then this hypothesis will be true as soon as $\mu_\theta$ has a density with respect to the Lebesgue measure.
>
> For this reason, except in particular situations (see our response to Reviewer GVxx "About the motivational example of Proposition 2"), one cannot expect a practical pitfall from the gradient problem that we highlighted in this paper.
>
> We agree to add remarks to discuss these regularity conditions in more details. In particular, we will emphasize on the fact that the hypotheses of the Theorems will be satisfied in most situations encountered in WGAN learning with $C^1$ generators.
>
> We will also better explain how these conditions can be verified in practice. The regularity condition for the generator is easily satisfied with a generator that has Lipschitz $C^1$ activation functions. Also, the x-regularity condition of the cost $c$ can be easily checked with the expression of $c$ (it is clearly true with the quadratic cost).

---

### Author Response · Authors · 2022-11-09
**Message for all reviewers**

We thank all reviewers for their careful reading and their helpful comments. We will soon post on OpenReview an updated version of the paper. Before that, we answer the reviewers' comments, and we explain how the reviewers' suggestions will be addressed in the updated submission.

---

> ### Author Response · Authors · 2022-11-13
> **Revised version of our paper**
>
> We have have just uploaded a new version of our submitted paper, following the requests expressed by the reviewers. Changes are indicated in red, except for complete new Sections where only the title is written in red.
>
> We have included a new Subsection 6.2 (with experiments on synthetic low-dimensional datasets) which makes the connection between our theoretical study and the practical behavior of the considered WGAN learning algorithm. These new experiments clarify why the gradient pitfall is generally avoided in practice, and also exhibit that the stability of the alternate algorithm is very dependent on the choice of hyperparameters of the optimization strategy (e.g. learning rates). The various convergence behaviors observed in these additional experiments provide some insights on the results obtained by WGAN learning on larger datasets (such as stability and mode-collapse), as showcased here for the MNIST dataset. We hope that this new section answers the main concern expressed by the reviewers about the disconnection between our experiments and our theoretical findings.
>
> We also added some detailed context throughout the paper to highlight the contributions of this paper with respect to existing works (three new paragraphs at the end of Section 1.2 "Related Works", more details in Section 1.3 "Contributions and outline", and several new remarks in Section 3 about the semi-discrete setting). In particular, in Remarks 7,8 at the end of Section 3, we now give more intuition on the results obtained in Section 3 and their consequence on practical WGAN learning. The interpretation based on semi-discrete OT (with Laguerre diagrams which are now illustrated in several new figures) helps to better understand the impact of using a sub-optimal dual variable when updating the generator.
>
> Finally, we added a brief discussion on the regularity hypotheses that we introduced for the cost and the generator. We provided some arguments to show that these hypotheses are very often satisfied in practical WGAN learning with $C^1$ generative neural networks.
>
> Here is the list of the major changes included in this new version:
> - a new Section 6.2 (and new Figures 4, 5, 6, and Figures 11, 12 in new Appendix E) with experiments on synthetic low-dimensional datasets (in particular, the 2-point dataset related to the counterexample of Propostion 2)
> - three new paragraphs at the end of Section 1.2 "Related works"
> - a detailed list of our main contribution in Section 1.3
> - several references to existing works on semi-discrete OT in Section 3
> - a new Figure 1 that illustrates the counterexample of Proposition 2
> - a new Figure 2 that illustrates the concepts of Laguerre diagram and semi-discrete transport map
> - a new paragraph after Definition 1 to discuss the $x$-regularity hypothesis on the ground cost $c$
> - a new paragraph just before Section 2.1 to clarify our notations for gradients and derivatives
> - a new Remark 1 to clarify the definition of the $c$-transform and its dependence on the chosen set $\mathcal{Y}$
> - a new Remark 3 to discuss the regularity hypothesis for the generator
> - a new Remark 4 to treat the example of Proposition 2 in the regularized case $\lambda > 0$
> - a new Remark 7 on the impact of using a sub-optimal dual variable during the update of the generator
> - a new Remark 8 that gives an insight on the connection between OT and WGAN learning
> - a new Appendix A that lists the main notations of the paper
> - a new Appendix C that includes all the details of our experimental setting
> - a new Appendix D that includes two additional experiments complementing the experimental analysis conducted in section 6.2
> - updated abstracts and conclusion
>
> Again, we would like to warmly thank all the reviewers for their many helpful suggestions.

---

### Decision · Action_Editors · 2023-02-02

**Recommendation:** Accept with minor revision

**Comment:**

I think the message of the paper is great. There are several interesting theorems that fill a corner of the literature. My only concern is that the paper size has grown too much after revisions. While this is great to include reviewers' comments, this has the side-effect of diluting further the message of the paper. It feels at times the paper holds a "pot-pourri" of several loosely connected results and remarks. Unfortunately, the criticism of a lack of a unified message was one of the issues highlighted by reviewers.

I also second the reviewers on the relative opacity of what's really brought forward by the experiments. They are all on fairly simple, if not synthetic data, and do not necessarily align with the rest of the paper. The abstract also concludes on a bias issue in experiments using Sinkhorn, but, on the other hand, debiasing is presented as a result in the theory part.

I think it is therefore important that the authors go back to the drawing board and significantly cut the length of the paper. I think going down to 20 pages for the main body would be reasonable, given that the main message of the paper is quite easy to parse. I really believe some areas are worth cutting, or redesigning completely, starting with Figures 3,4,5,6 in the experiment that are barely readable, and all other figures which I find overall poorly executed (legends or axis ticks are not readable). In terms of content, it's not entirely clear to me what Section 5 provides, and the abundance of numbered Remarks does blur the overall message. To what extent should a remark be part of the main message of the paper, rather than a technical side-effect of a result that is proved? Putting them on equal footing makes it quite bad to analyze.

I think that, while I expect items above will require significant work to improve presenation, they are "only" cosmetic, and will mostly involve making editorial choices, which is why I am inclined to accept pending a (not so minor) revision on form.

minor comment:
- "repercuted" (Rk.8) is not an English word

**Audience:**

Differentiating OT problems is now very common. Here the authors focus on differentiating min's (the objective), whereas other works have taken interest in differentiating the argmin's (the OT coupling, potentials, or maps). There will be an audience interested in using these more rigorous theoretical results to support other algorithmic approaches.

**Claims And Evidence:**

This paper proposes an in-depth study of issues arising when differentiating OT w.r.t. to a parameterized measure. The paper argues that the Danskin theorem, while intuitive and often applied in that context, does not always necessarily hold. While the "failure" case where such issues arise is arguably a corner case (Prop.2), that result is interesting in itself. The authors then examine more closely various common scenarios, such as semi-discrete (unregularized), Sinkhorn regularized OT cost and Sinkhorn divergence in various set. They follow with generation experiments.